# BAPO: Stabilizing Off-Policy Reinforcement Learning for LLMs via Balanced Policy Optimization with Adaptive Clipping

**Zhiheng Xi**[1][*][†], **Xin Guo**[1][*], **Yang Nan**[1], **Enyu Zhou**[1], **Junrui Shen**[1],
**Wenxiang Chen**[1], **Jiaqi Liu**[1], **Jixuan Huang**[1], **Zhihao Zhang**[1], **Honglin Guo**[1],
**Xun Deng**[2], **Zhikai Lei**[2], **Miao Zheng**[2], **Guoteng Wang**[2], **Shuo Zhang**[2], **Peng Sun**[2],
**Rui Zheng**[2], **Hang Yan**[2], **Tao Gui**[1,3][†], **Qi Zhang**[1][†], **Xuanjing Huang**[1]

[1]Fudan University [2]Shanghai Qiji Zhifeng Co., Ltd. [3]Shanghai Innovation Institute
zhxi22@m.fudan.edu.cn, {tgui,qz}@fudan.edu.cn

## Abstract

Reinforcement learning (RL) has recently become the core paradigm for aligning and strengthening large language models (LLMs). Yet, applying RL in off-policy settings—where stale data from past policies are used for training—improves sample efficiency, but remains challenging: policy entropy declines sharply, optimization often becomes unstable and may even collapse. Through theoretical and empirical analysis, we identify two key insights: (i) **an imbalance in optimization**, where negative-advantage samples dominate the policy gradient, suppressing useful behaviors and risking gradient explosions; and (ii) **the derived Entropy-Clip Rule**, which reveals that the fixed clipping mechanism in PPO-like objectives systematically blocks entropy-increasing updates, thereby driving the policy toward over-exploitation at the expense of exploration. Building on these insights, we propose **BA**lanced **P**olicy **O**ptimization with Adaptive Clipping (**BAPO**), a simple yet effective method that dynamically adjusts clipping bounds to adaptively re-balance positive and negative contributions, preserve entropy, and stabilize RL optimization. Across diverse off-policy scenarios—including sample replay and partial rollout—BAPO achieves fast, stable, and data-efficient training. On AIME 2024 and AIME 2025 benchmarks, our 7B BAPO model surpasses open-source counterparts such as SkyWork-OR1-7B, while our 32B BAPO model not only achieves state-of-the-art results among models of the same scale but also outperforms leading proprietary systems like o3-mini and Gemini-2.5-Flash-Thinking.

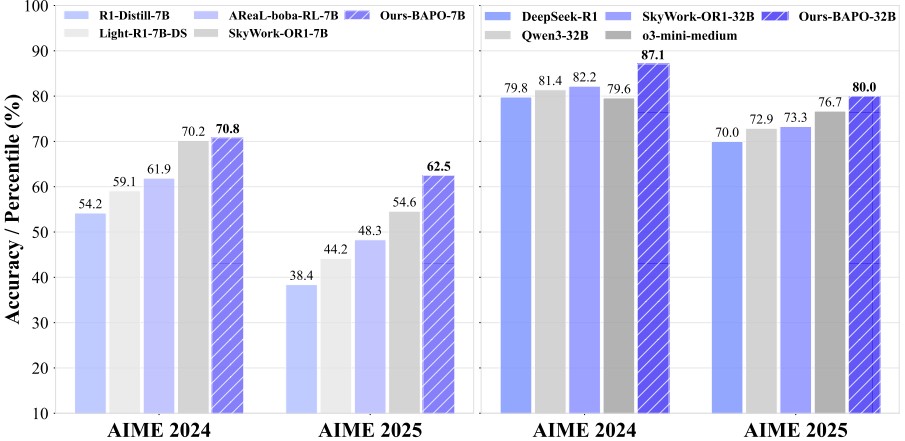

Figure 1: Performance of BAlanced Policy Optimization with Adaptive Clipping (BAPO).

---

[*]Equal contribution.[†]Corresponding authors.

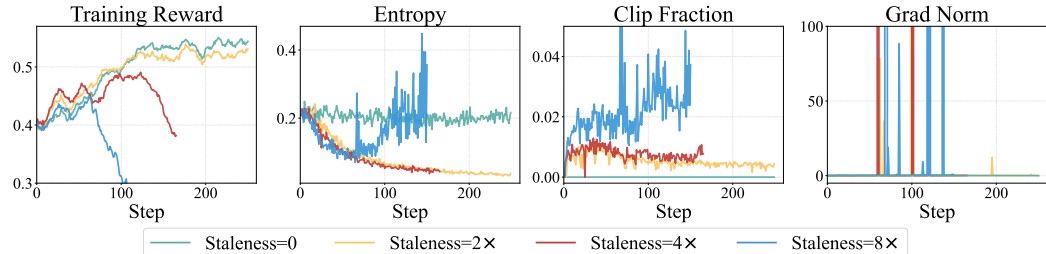

Figure 2: Preliminary results with different data staleness. As the staleness increases, the model suffers from unstable optimization, decreasing entropy, and even a sudden collapse in training.

# 1 INTRODUCTION

Reinforcement learning (RL) has become a pivotal paradigm for optimizing large language models (LLMs) (Zhang et al., 2025), delivering significant improvements in complex tasks such as reasoning (Jaech et al., 2024; Guo et al., 2025a), coding (Anthropic, 2025), and agentic decision-making (Bai et al., 2025; Xi et al., 2025). Among RL methods, off-policy RL—where the rollout policy (behavior policy) differs from the training policy (target policy)—emerges as particularly promising (Roux et al., 2025b; Arnal et al., 2025). It offers high sample efficiency and tolerance to data staleness, making it well-suited for extremely long-horizon and challenging scenarios, while also aligning more naturally with features in modern AI infrastructures such as partial rollout (Team et al., 2025; Fu et al., 2025).

However, applying off-policy RL to LLMs introduces substantial challenges (Yu et al., 2025; Arnal et al., 2025). As shown in Figure 2, increasing data staleness leads to unstable optimization, exploding gradient and even collapse. Meanwhile, policy entropy declines sharply, reflecting reduced exploratory capacity and a bias toward over-exploitation. By contrast, on-policy training—where rollout and target policies coincide—remains stable across metrics, consistent with prior studies (Tang et al., 2024; Roux et al., 2025b; Arnal et al., 2025).

To understand the instability of off-policy training, we conduct a comprehensive theoretical and empirical analysis to reveal two key insights. We first demonstrate an **imbalance in optimization**: policy updates are often dominated by negative-advantage samples, producing excessive penalization signals that suppress even neutral or correct actions and may cause gradient explosions (Gülçehre et al., 2023). We then derive and empirically validate the **Entropy-Clip Rule** in the widely-used PPO (Schulman et al., 2017) and GRPO (Shao et al., 2024), showing that the clipping mechanism in PPO-like objectives blocks many low-probability positive tokens while over-penalizing low-probability negatives. This systematically excludes entropy-increasing updates, sharpens the output distribution, and drives policies toward over-exploitation at the cost of exploration.

Based on these insights, we propose **BA**lanced **P**olicy **O**ptimization with Adaptive Clipping (**BAPO**), a new method for stable and effective off-policy RL. BAPO dynamically adjusts the clipping bounds to re-balance positive and negative contributions for each update step, incorporate low-probability positives while filtering excessive negatives, and preserve policy entropy—achieving a better balance between exploration and exploitation. An overview of our approach is illustrated on the right side of Figure 3.

Experiments across diverse off-policy scenarios—including sample replay, partial rollout, and varying degrees of staleness—on base models such as DeepSeek-R1-Distill-Qwen-7B (Guo et al., 2025a) and OctoThinker-Llama3.2-3B-Long-Zero (Wang et al., 2025b) show that BAPO consistently yields significant improvements. Our 7B model achieves scores of 70.8 on AIME24 and 62.5 on AIME25, surpassing open-source counterparts such as SkyWork-OR1-7B (He et al., 2025). Moreover, our 32B model reaches 87.1 on AIME24 and 80.0 on AIME25, outperforming both comparably scaled open-source models like Qwen3-32B (Yang et al., 2025a) and leading proprietary systems including o3-mini-medium (OpenAI, 2025) and Gemini-2.5-Flash-Thinking (Comanici et al., 2025).

Our contributions are summarized as follows:

- We identify and analyze two key insights behind instability in off-policy RL for LLMs: the imbalanced optimization and the Entropy-Clip Rule. (§3)

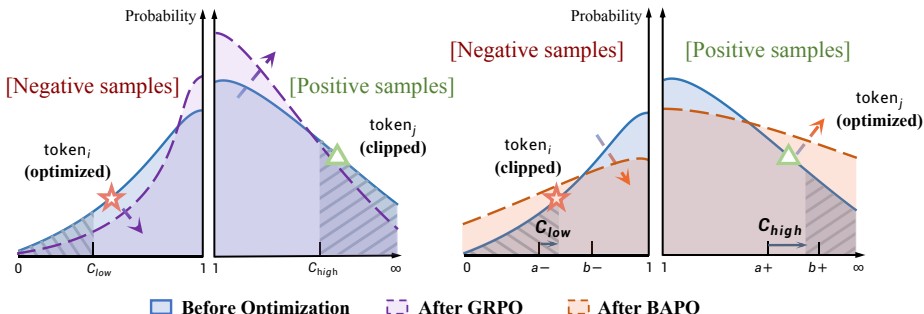

Figure 3: An illustration of our proposed BAPO. **(Left)** Baseline methods like GRPO use symmetric fixed clipping bounds, reinforcing high-probability positive tokens while penalizing excessive low-probability negatives, leading to sharp distributions and entropy collapse. **(Right)** BAPO dynamically adjusts the clipping bounds $c_{\text{low}}$ and $c_{\text{high}}$ based on the loss contributions from positive tokens. It excludes overly negative tokens ⭐ to maintain a smoother distribution and incorporates previously clipped positive tokens △ to preserve entropy balance.

- We propose BAPO, a new RL algorithm that dynamically adjusts clipping bounds to balance positive and negative signals, preserving entropy for exploration, and stabilizing training. (§4)
- We validate BAPO across multiple backbones, model scales, and off-policy settings, showing that it achieves stable optimization and competitive results with proprietary systems.[1] (§5)

## 2 PRELIMINARIES

**Policy gradient.** In the field of LLM RL (Trung et al., 2024; Jaech et al., 2024; Xi et al., 2024b), policy gradient-based (PG) algorithms (Williams, 1992) are widely used. Specifically, given an input prompt $\boldsymbol{x}$, an LLM $\pi_\theta$ sequentially generates a $T$-token response $\boldsymbol{y} = (y_1, ..., y_T)$:

$$\pi_\theta(\boldsymbol{y}|\boldsymbol{x}) = \prod_{t=1}^T \pi_\theta(y_t|\boldsymbol{x}, \boldsymbol{y}_{<t}) . \tag{1}$$

Given a training dataset $\mathcal{D} = \{\boldsymbol{x}_1, ..., \boldsymbol{x}_N\}$ and reward function $R$, the RL objective is to maximize the expected reward:

$$J(\theta) = \mathbb{E}_{\boldsymbol{x}\sim\mathcal{D},\ \boldsymbol{y}\sim\pi_\theta(\cdot|\boldsymbol{x})} [R(\boldsymbol{x}, \boldsymbol{y})] . \tag{2}$$

PG algorithms then leverage gradient ascent to optimize the policy with the following gradient:

$$\nabla_\theta J(\theta) = \mathbb{E}_{\boldsymbol{x}\sim\mathcal{D},\ \boldsymbol{y}\sim\pi_\theta(\cdot|\boldsymbol{x})} \left[ \sum_{t=1}^T \nabla_\theta \log \pi_\theta(y_t|\boldsymbol{x}, \boldsymbol{y}_{<t}) \cdot A_t \right] , \tag{3}$$

where $A_t$ denotes the estimated advantage at time step $t$, i.e., how much better action $y_t$ is than the expected action under the current policy.

**Importance sampling and PPO objective.** To improve sample efficiency and adapt to modern infrastructure, mainstream RL algorithms for LLMs typically adopt a PPO-like surrogate objective (Schulman et al., 2017):

$$J^{\text{PPO}}(\theta) = \mathbb{E}_{\boldsymbol{x}\sim\mathcal{D},\ \boldsymbol{y}\sim\pi_{\theta_{\text{rollout}}}(\cdot|\boldsymbol{x})} \sum_{t=1}^T \left[ \min(r_t \cdot A_t, \text{clip}(r_t, 1-\varepsilon, 1+\varepsilon) \cdot A_t) \right] , \tag{4}$$

where $r_t = \frac{\pi_\theta(y_t|\boldsymbol{x}, \boldsymbol{y}_{<t})}{\pi_{\theta_{\text{rollout}}}(y_t|\boldsymbol{x}, \boldsymbol{y}_{<t})}$ is the importance weight that corrects for the distribution mismatch, estimating the expected advantage of tokens generated by the behavior policy $\pi_{\theta_{\text{rollout}}}$ under the target policy $\pi_\theta$. The clipping mechanism in PPO serves to implicitly enforce a trust region between the

---

[1]Our code are available at `https://github.com/WooooDyy/BAPO`.

Figure 4: Contribution of positive and negative tokens to the policy-gradient loss and their proportion of tokens during training.

Figure 5: Relationship between token probability and importance sampling weight.

behavior and target policies, preventing overly large policy updates that could destabilize training. The hyperparameter $\varepsilon \in (0, 1)$ determines the width of this clipping interval.

We then analyze data with positive and negative advantages respectively. The policy gradient can then be expressed as:

$$\nabla J^{\text{PPO}} = \sum_{\boldsymbol{x}, y_t} [\underbrace{\mathbb{I}(A_t > 0) \cdot \mathbb{I}\{r_t < 1 + \varepsilon\} \cdot A_t \cdot \pi_\theta(y_t) \cdot \nabla \log \pi_\theta(y_t)}_{\text{positive tokens}} \\ + \underbrace{\mathbb{I}(A_t < 0) \cdot \mathbb{I}\{r_t > 1 - \varepsilon\} \cdot A_t \cdot \pi_\theta(y_t) \cdot \nabla \log \pi_\theta(y_t)}_{\text{negative tokens}}], \tag{5}$$

where $\mathbb{I}$ represents the indicator function.

## 3 MOTIVATION: IMBALANCED OPTIMIZATION AND ENTROPY-CLIP RULE

In this section, we first conduct preliminary experiments to show the influence of data staleness on the RL optimization process. Next, we perform in-depth empirical and theoretical analysis to reveal the underlying mechanisms and provide new insights.

**Training instability with data staleness.** We perform experiments under different levels of data staleness using the popular GRPO algorithm. Results in Figure 2 show that, compared to on-policy training, off-policy RL typically suffers from instability, and entropy decreases rapidly, reflecting reduced exploratory capacity (He et al., 2025). As staleness increases, the entropy decline becomes more severe and a larger number of tokens are clipped; meanwhile, training becomes more unstable. In the following paragraphs, we attempt to explain this phenomenon from different perspectives and summarize the motivation behind our method.

**Excessive negative samples lead to imbalanced optimization.** Within the PPO-like objective for policy updates, we analyze tokens with positive and negative advantages separately, as shown in Equation 5. Empirical results in Figure 4 reveal a pronounced imbalance: positive samples constitute a minority both in number and in their contribution to the policy-gradient loss. We attribute this skew to two main factors: (i) the model tends to generate longer trajectories on difficult queries, thereby producing more tokens in negative samples (Figure 6); and (ii) in early stages of training, the model has not yet acquired sufficient capability, resulting in a higher proportion of negative samples. This observation may help explain the effectiveness of certain curriculum-based approaches (Xi et al., 2024a; Yuan et al., 2025).

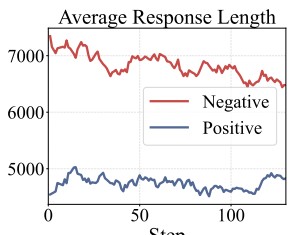

Figure 6: Average model response length during training.

In the RL training of LLMs, reinforcing positive samples is often more efficient for driving performance gains than attempting to "suppress" the vast number of negative samples (Gülçehre et al., 2023; Zhu et al., 2025). To this end, prior work has proposed amplifying positive signals through the clip-higher technique (Yu et al., 2025). However, merely enlarging the clipping upper bound does not mitigate the influence of negative data, thus failing to prevent them from dominating the optimization process. Moreover, as shown in Equation 5, the accumulation of low-probability negative

tokens (i.e., $\pi_\theta(y_t) \to 0$, driving the log term toward $-\infty$) may trigger gradient explosion, further destabilizing training (Yang et al., 2025c).

**The Entropy-Clip Rule exposes insufficient entropy promotion in optimization, leading to entropy collapse.** Theoretically, we derives Equation 6 (see Appendix G for detailed derivations) for PPO surrogate objective to reveal the factors that influence the policy entropy (Roux et al., 2025b):

$$\Delta\mathcal{H}(\pi_\theta) \approx -\eta \cdot \text{Cov}_{\boldsymbol{y} \sim \pi_\theta(\cdot|\boldsymbol{x})} \left[\log \pi_\theta(y_t|\boldsymbol{x}, \boldsymbol{y}_{<t}), A_t \cdot \mathcal{X}(y_t) + C\right] , \qquad (6)$$

where $C$ is a constant, and

$$\mathcal{X}(y_t) = \begin{cases} 1, & \textbf{if } A_t > 0 \ \& \ r_t < 1 + \epsilon \\ & \textbf{or } A_t < 0 \ \& \ r_t > 1 - \epsilon \\ 0, & \text{otherwise.} \end{cases} \qquad (7)$$

We observe that changes in policy entropy are driven by the influence of unclipped tokens, which is determined by the covariance between their log probabilities and advantages. We term this as **the Entropy-Clip Rule**. The left side of Figure 3 illustrates how the optimization of different tokens influences the probability distribution, thereby affecting entropy. The Entropy-Clip Rule theoretically explains the following statement: Specifically, updating the policy with positive high-probability tokens (high advantage, high probability) and negative low-probability tokens (low advantage, low probability) sharpens the distribution and consequently reduces entropy. Conversely, updating the policy with negative high-probability tokens and positive low-probability tokens smooths the distribution, resulting in an increase in entropy (detailed proofs are available in Appendix G.4.2).

Empirically, our statistical analysis on token probabilities and their importance sampling (IS) weights further clarifies this dynamic. As shown in Figure 5, we find that tokens with either very high or very low IS weights tend to have low probabilities. However, in standard algorithms with symmetric clipping bounds (e.g., [0.8,1.2]), a majority of positive, low-probability tokens are prevented from contributing to the optimization. This systematic exclusion of entropy-increasing updates causes a continuous decline in entropy, ultimately crippling the model's exploratory capacity and resulting in a performance bottleneck.

**Summary of motivation.** Based on the above analysis, we can summarize two main motivations: (1) to balance the contributions of positive and negative tokens while preventing gradient explosion, and (2) to preserve policy entropy for sustaining exploration and preventing collapse.

## 4 METHODOLOGY

### 4.1 VALIDATION EXPERIMENT: ASYMMETRIC CLIPPING

The main idea of our method is to stabilize the training and maintain exploration ability of the policy by asymmetrically adjusting the trust region for positive and negative tokens, i.e., adjusting $c_{\text{low}}$ and $c_{\text{high}}$.

We then conduct preliminary validation experiments to examine whether asymmetrically adjusting the clipping range could influence the training dynamics. The results, shown in Figure 7, together with Figure 5, reveal that increasing the upper bound $c_{\text{high}}$ (which introduces more low-probability positive tokens to policy updates) improves performance while counteract-

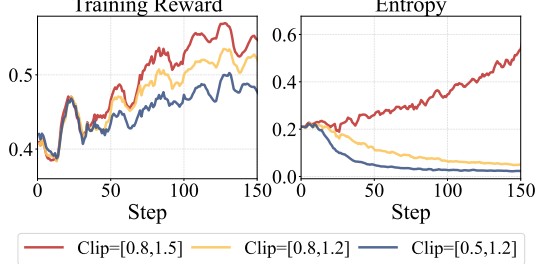

Figure 7: Training dynamics of asymmetric clipping experiments.

ing the downward trend of entropy, albeit at a rapid pace. In contrast, relaxing the lower bound $c_{\text{low}}$ (which introduces more low-probability negative tokens to policy updates) not only degrades

---

**Algorithm 1:** BAPO

---

**Input:** Initialized LLM policy $\pi_\theta$, training dataset $\mathcal{D}$, reward function $R$, staleness $E$, movable range of clipping bounds $[a^-, b^-]$ and $[a^+, b^+]$, step size of upper bound $\delta_1$, step size of lower bound $\delta_2$, positive token contribution threshold $\rho_0$

1 **for** *step $s = 1...S$* **do**
2      **Procedure** Sample and filter out responses
3          Update the old LLM policy $\pi_{\theta_{\text{rollout}}} \leftarrow \pi_\theta$ ;
4          Sample the $s$-th batch $\mathcal{D}_s$ from $\mathcal{D}$ ;
5          Sample $G$ responses $\{\boldsymbol{y}_i\}_{i=1}^{G} \sim \pi_{\theta_{\text{rollout}}}(\cdot|\boldsymbol{x})$, where $\boldsymbol{x} \in \mathcal{D}_s$ ;
6          Compute reward and advantage for each $\boldsymbol{y}_i$ based on reward function $R$ ;
7      **for** *staleness $= 0...E$* **do**
8          **Procedure** Dynamically adjusting the clipping bounds $c_{\text{high}}$ *and* $c_{\text{low}}$
9              Initialize clipping bounds $c_{\text{low}} = a^-$ and $c_{\text{high}} = a^+$ ;
10              **while** *the positive token contribution $\rho < \rho_0$* **and** $c_{\text{low}} + \delta_2 \leq b^-$
11              **do**
12                  **if** $c_{\text{high}} + \delta_1 \leq b^+$ **then**
13                      $c_{\text{high}} \leftarrow c_{\text{high}} + \delta_1$
14                  **else**
15                      $c_{\text{low}} \leftarrow c_{\text{low}} + \delta_2$
16                  **end**
17              **end**
18          **Procedure** Update the LLM policy $\pi_\theta$
19              Update the LLM policy $\pi_\theta$ by maximizing the following objective:
20              $J^{\text{BAPO}}(\theta) = \mathbb{E}_{\boldsymbol{y} \sim \pi_{\theta_{\text{rollout}}}(\cdot|\boldsymbol{x})} \sum_{t=1}^{T} \left[ \min(r_t \cdot A_t, \text{clip}(r_t, c_{\text{low}}, c_{\text{high}}) \cdot A_t) \right]$
21      **end**
22 **end**

---

performance but also accelerates entropy collapse. These findings confirm the effectiveness of entropy control through asymmetric clipping. Nevertheless, this approach remains relatively rigid and manually specified, providing limited flexibility and adaptation.

## 4.2 BAPO: BALANCED POLICY OPTIMIZATION WITH ADAPTIVE CLIPPING

To this end, we propose **BA**lanced **P**olicy **O**ptimization with Adaptive Clipping (**BAPO**), a new method to achieve stable, fast RL optimization for LLMs. The core insight of BAPO lies in its adaptive clipping mechanism, which dynamically adjusts the clipping bounds $c_{\text{high}}$ and $c_{\text{low}}$, to regulate the positive contribution to the policy loss and maintain a balance in entropy throughout RL training. Formally, for each update with a batch, our goal is to find a pair of $c_{\text{high}}$ and $c_{\text{low}}$ that satisfy:

$$\frac{\sum_{A_t > 0} \pi_{\theta_{\text{rollout}}}(y_t) \cdot |\min(r_t \cdot A_t, \text{clip}(r_t, 0, c_{\text{high}}) \cdot A_t)|}{\sum_{A_t} \pi_{\theta_{\text{rollout}}}(y_t) \cdot |\min(r_t \cdot A_t, \text{clip}(r_t, c_{\text{low}}, c_{\text{high}}) \cdot A_t)|} \geq \rho_0 \, , \tag{8}$$

where $\rho_0$ is the target contribution of positive signals to the policy gradient loss. Specifically, BAPO gradually increases $c_{\text{high}}$ and $c_{\text{low}}$ with step sizes of $\delta_1$ and $\delta_2$, respectively, until the condition in Equation 8 is met. We present an overview of BAPO in Figure 3 and summarize it in Algorithm 1.

Overall, BAPO offers several significant benefits. First, by dynamically adjusting $c_{\text{high}}$ and $c_{\text{low}}$ for each step, we can increase the contribution of positive tokens to the policy-gradient loss while preventing negative tokens from excessively dominating the optimization objective. Second, based on our earlier analysis of the relationship between IS weights and token probabilities in Figure 5, BAPO incorporates more low-probability positive tokens and filters out more low-probability negative tokens, both of which contribute to maintaining entropy. Third, by setting the target contribution from positive tokens, BAPO prevents uncontrolled entropy growth, avoids situations where positive tokens overwhelm the loss, and mitigates tail degradation—where the model overfits to easy problems but fails to handle more challenging ones (Ding et al., 2025b).

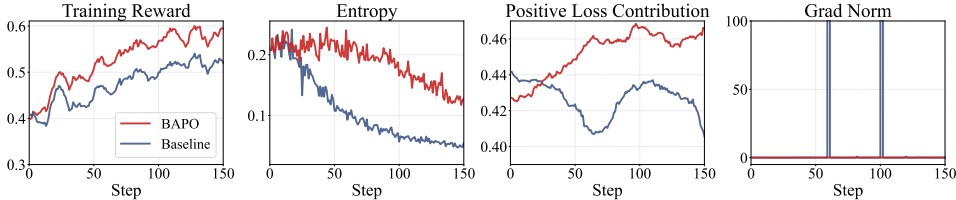

Figure 8: Training dynamics of BAPO.

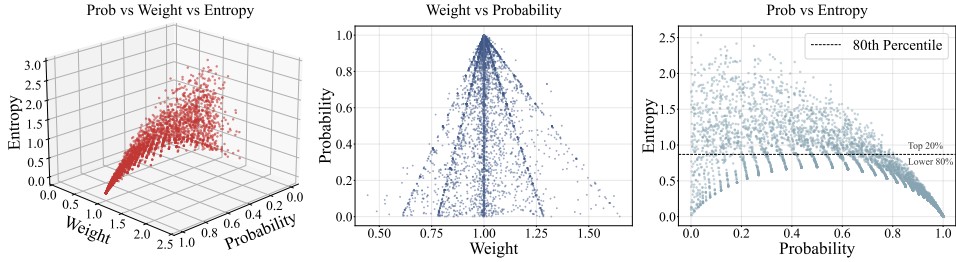

Figure 9: Relationship among token probabilities, importance sampling weights, and entropy.

## 4.3 ANALYSIS

**Stable and fast training of BAPO.** As shown in Figure 8, BAPO enables a more stable optimization process, characterized by rapidly increasing training rewards, greater contributions from positive tokens, steady gradient normalization, and stable policy entropy—resulting in an improved balance between exploration and exploitation.

We further visualize the adjustment process of the clipping bounds in BAPO. As shown in Figure 10, the averaged upper and lower clipping bounds both fluctuate during training, confirming that BAPO dynamically adjusts the clipping for both types of data and adaptively balances their contributions to the loss. In contrast to approaches such as DAPO (Yu et al., 2025) or the asymmetric clipping in Section 4.1, which rely on empirical tuning, BAPO eliminates the need for complex manual hyperparameter tuning, making it simple yet effective.

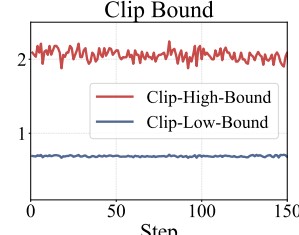

Figure 10: Clipping bounds.

**Effectiveness of BAPO across different staleness.** We conduct experiments using the R1-Distill model (Guo et al., 2025a) on the SkyWork-OR1-RL dataset (He et al., 2025), with a maximum sequence length of $32k$. The results in Figure 11 show that under different data staleness, our method consistently outperforms both the baseline and the clip-higher approach, demonstrating its superiority.

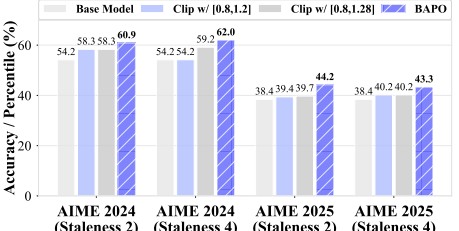

Figure 11: Results with different data staleness.

**The working mechanism of BAPO and its connection to prior work.** To better understand the working mechanism of BAPO, we present the relationship among token probabilities, IS weights, and entropy during training in Figure 9. We find that as IS weights deviate further from 1, the corresponding token probabilities decrease, and such low-probability tokens often exhibit higher entropy. Based on this observation, we explain how BAPO relates to prior work. For example, Clip-Higher in Yu et al. (2025) sets the clipping upper bound to $1.28$, thereby including more low-probability positive tokens in training, which stabilizes entropy while balancing the contributions of positive and negative tokens. Similarly, Wang et al. (2025a) retain only the top 20% highest-entropy tokens for training, ensuring stable entropy throughout optimization and preserving the model's

Table 1: Main evaluation results.

| Model | Model Size | AIME 2024 | AIME 2025 | Average |
|---|---|---|---|---|
| ≥ 100B Models and Proprietary Models | | | | |
| Qwen3-235B-A22B (Yang et al., 2025a) | 235B | 85.7 | 81.5 | 83.6 |
| DeepSeek-R1 (Guo et al., 2025a) | 671B | 79.8 | 70.0 | 74.9 |
| DeepSeek-R1-0528 (Guo et al., 2025a) | 671B | 91.4 | 87.5 | 89.5 |
| o1$_{medium}$ (Jaech et al., 2024) | - | 83.3 | 79.0 | 81.2 |
| o3-mini$_{medium}$ (OpenAI, 2025) | - | 79.6 | 76.7 | 78.2 |
| o3-mini$_{high}$ (OpenAI, 2025) | - | 87.3 | 86.5 | 86.9 |
| Gemini-2.0$_{Flash-Thinking}$ (Google, 2024) | - | 73.3 | 53.5 | 63.4 |
| Gemini-2.5$_{Flash-Thinking-0520}$ (Comanici et al., 2025) | - | 82.3 | 72.0 | 77.2 |
| 10B - 100B Scale Models | | | | |
| Qwen3-30B-A3B (Yang et al., 2025a) | 30B | − | 61.3 | − |
| R1-Distill-Qwen-32B (Guo et al., 2025a) | 32B | 72.6 | 54.9 | 63.8 |
| QwQ-32B (Qwen, 2025) | 32B | 79.5 | 65.3 | 72.4 |
| Qwen3-32B (Yang et al., 2025a) | 32B | 81.4 | 72.9 | 77.2 |
| SkyWork-OR1-32B (He et al., 2025) | 32B | 82.2 | 73.3 | 77.8 |
| BP-Math-32B$_{SFT}$ | 32B | 84.4 | 78.1 | 81.3 |
| BP-Math-32B$_{GRPO}$ | 32B | 84.6 | 78.8 | 81.7 |
| BP-Math-32B$_{BAPO}$ | 32B | **87.1** | **80.0** | **83.5** |
| ≤ 10B Models | | | | |
| R1-Distill-Qwen-7B (Guo et al., 2025a) | 7B | 54.2 | 38.4 | 46.3 |
| Light-R1-7B-DS (Wen et al., 2025) | 7B | 59.1 | 44.2 | 51.7 |
| AReaL-boba-RL-7B (Fu et al., 2025) | 7B | 61.9 | 48.3 | 55.1 |
| AceReason-Nemotron-7B (Chen et al., 2025c) | 7B | 69.0 | 53.6 | 61.3 |
| SkyWork-OR1-7B (He et al., 2025) | 7B | 70.2 | 54.6 | 62.4 |
| BP-Math-7B$_{SFT}$ | 7B | 66.9 | 59.0 | 62.9 |
| BP-Math-7B$_{GRPO}$ | 7B | 69.2 | 59.2 | 64.2 |
| BP-Math-7B$_{BAPO}$ | 7B | **70.8** | **62.5** | **66.7** |

exploratory capability, and the target entropy technique in He et al. (2025) plays a similar role, which aligns with our motivation.

## 5 EXPERIMENTS AND DISCUSSION

### 5.1 EXPERIMENTAL SETUPS

**Datasets and Models.** We use SkyWork-OR1-RL-Data (He et al., 2025) as our RL dataset, as it is widely adopted and of high quality. For evaluation, we employ both the AIME 2024 and the newly released AIME 2025 (AIME, 2025) benchmarks. Our experiments cover a range of backbone models, including DeepSeek-R1-Distill-Qwen-7B, DeepSeek-R1-Distill-Qwen-32B (Guo et al., 2025a), and OctoThinker-Llama3.2-3B-Long-Zero (Wang et al., 2025b). In addition, we incorporate two our own supervised fine-tuning (SFT) models, BP-Math-7B and BP-Math-32B, which are derived from Qwen2.5-Math (Yang et al., 2024) through fine-tuning.

**Implementation details.** We leverage GRPO as the basis for BAPO. Both our preliminary and validation experiments are conducted using DeepSeek-R1-Distill-Qwen-7B, with the maximum response length set to $8k$, learning rate to $2 \times 10^{-6}$, and temperature to $0.6$. For main results on BP-Math models, we set the maximum response length to $64k$ to align with the SFT setting. To introduce staleness, we adopt multiple strategies, including experience reuse through ppo_epoch (Schulman et al., 2017) and the modern partial rollouts (Team et al., 2025; Fu et al., 2025). For BAPO, we set the target contribution $\rho_0 = 0.4$, the movable range $a^- = 0.6$, $b^- = 0.9$, $a^+ = 1.2$, $b^+ = 3.0$, and the step size $\delta_1 = 0.05$, $\delta_2 = 0.02$. These hyperparameters are not finely tuned, as they already demonstrate strong empirical performance. For evaluation, we report results averaged over 16 rollouts.

**Baselines.** We include a variety of commercial and open-source models of different scales as baselines, as shown in Table 1, and report their performance as extracted from prior work. In addition, we compare different training approaches, including SFT and GRPO.

### 5.2 MAIN RESULTS

The main results are shown in Figure 1 and Table 1.

**Significant performance improvements across models of varying sizes.** For strong SFT models, GRPO provides only marginal benefits—for instance, it improves performance by just 0.2 and 0.7 points on AIME24 and AIME25 with the `BP-Math-32B` model. In contrast, BAPO delivers substantial gains across models of different scales. Specifically, with the `BP-Math-32B` model, BAPO outperforms SFT by 2.7 and 1.9 points on AIME24 and AIME25, respectively; with the `BP-Math-7B` model, it achieves even larger improvements of 3.9 and 3.5 points.

**SOTA performance over open-source models of comparable sizes and competitive results against proprietary models.** Compared to open-source models of similar sizes, our BAPO-trained models achieve state-of-the-art (SOTA) performance. For instance, among 32B models, `BP-Math-32B_BAPO` outperforms `Qwen3-32B` by 5.7 and 7.1 points on AIME24 and AIME25, respectively, and surpasses `SkyWork-OR1-32B` by 4.9 and 6.7 points. Among 7B models, `BP-Math-7B_BAPO` also delivers a notable 7.9-point improvement over `SkyWork-OR1-7B` on AIME25.

Moreover, `BP-Math-32B_BAPO` even outperforms some larger-scale models—for example, it surpasses `DeepSeek-R1` by 7.3 and 10.0 points on AIME24 and AIME25, respectively—while achieving performance comparable to `o3-mini`. Notably, even the smaller `BP-Math-7B_BAPO` yields results on par with `Gemini-2.0-Flash-Thinking`, underscoring the competitiveness of our approach against commercial models.

## 5.3 DISCUSSION

**Partial rollout.** To speed up rollouts in LLM reinforcement learning, modern AI infrastructures have introduced several techniques, with partial rollout being particularly noteworthy (Team et al., 2025; Fu et al., 2025). In this approach, long trajectories are split into segments: when a rollout exceeds a fixed token budget, the unfinished portion is stored in a replay buffer and resumed in later iterations instead of being regenerated from scratch. While this improves training efficiency, it also introduces off-policy

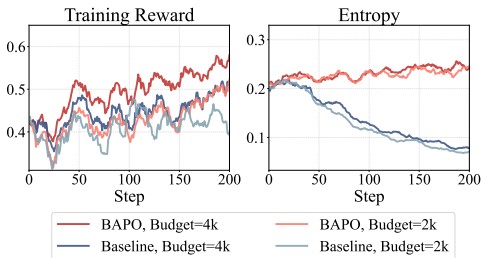

Figure 12: Training dynamics with partial rollout.

learning, since different parts of the same trajectory may come from multiple outdated policies. We evaluate BAPO under this setting, as shown in Figure 12. Compared to the baseline GRPO, BAPO exhibits greater robustness to such off-policy infrastructures and achieves more stable optimization.

**Results on OctoThinker-Llama3.2-3B-Long-Zero.** In addition to the DeepSeek-R1-Distill-Qwen, we also conducted experiments on Llama-based models (Wang et al., 2025b). As shown in Table 2 and Figure 13 in Appendix B, our method achieves more competitive results and exhibits greater stability in training dynamics.

Table 2: Performance of Llama-based models.

| Method | AIME 2024 | AIME 2025 | MATH |
|---|---|---|---|
| GRPO | 2.5% | 2.9% | 58.4% |
| BAPO | 5.4% | 5.8% | 66.0% |

**Generalization across diverse reasoning domains.** To further validate the universality and robustness of BAPO beyond standard mathematical benchmarks, we expanded our evaluation scope to cover a broader spectrum of reasoning tasks. Experiments were conducted using DeepSeek-R1-Distill-Qwen-7B. We configured the maximum sequence length to 16k for the logical reasoning task to accommodate long-context few-shot examples, while maintaining an 8k limit for mathematical and scientific tasks. As shown in Table 3 and Figure 14 in Appendix C, BAPO consistently delivers substantial performance gains across all tested domains.

**Comparison with other algorithms.** Beyond the standard GRPO baseline, we extended our evaluation to systematically compare BAPO with advanced variants sharing similar motivations, including TOPR Roux et al. (2025a), DAPO Yu et al. (2025), and DCPO Yang et al. (2025b). All methods were implemented and evaluated using the identical BP-Math-7B backbone and BAPO consistently outperforms these strong baselines as presented in Table 4. This superior performance validates that, among current advanced variants of GRPO, our adaptive clipping strategy offers the most effective solution for stabilizing training and maximizing reasoning capabilities.

Table 3: Comparison of performance across different domains. Results includes AMC 2023 and OlympiadBench He et al. (2024) for advanced mathematical reasoning, ARC-AGI Chollet (2019) for logical reasoning, and GPQA-Diamond Rein et al. (2024) for scientific question answering.

| Domain | Training Set | Test Set | Original | GRPO | BAPO | Improvement |
|--------|--------------|----------|----------|------|------|-------------|
| Math | Skywork Math | AMC 2023 | 73.3 | 82.5 | **92.5** | +19.2 |
| | | OlympiadBench | 33.6 | 54.2 | **58.4** | +25.4 |
| Logical | Enigmata Chen et al. (2025a) | ARC-AGI | 0 | 1.4 | **3.2** | +3.2 |
| General | General-Reasoner Ma et al. (2025) | GPQA-Diamond | 42.4 | 45.7 | **51.3** | +8.9 |

**Robustness to hyperparamteters**   To demonstrate the robustness of BAPO and assess its sensitivity to hyperparameter settings, we conducted comprehensive ablation studies on the BP-Math-7B model. We specifically examined three key components of our algorithm: (1) the positive token contribution threshold $\rho_0$, (2) the movable ranges of clipping bounds $[a_-, b_-]$ and $[a_+, b_+]$, and (3) the adjustment step sizes $\delta_1$ and $\delta_2$. The results are detailed in Appendix E. Overall these experiments demonstrate that BAPO does not require intricate hyperparameter tuning. Its key advantage lies in the ability to automatically adapt clipping values within broad search spaces, achieving superior performance using simple heuristic settings.

## 6 RELATED WORK

Recent landmark models, like OpenAI o1 (Jaech et al., 2024), DeepSeek-R1 (Guo et al., 2025a), Gemini 2.5 (Comanici et al., 2025), QwQ (Qwen, 2025), have demonstrated that reinforcement learning can effectively enable long chain-of-thought reasoning in LLMs (Shao et al., 2024; Zhang et al., 2025; Cai et al., 2025; Chen et al., 2025b). Mainstream algorithms include PPO (Schulman et al., 2017) and GRPO (Shao et al., 2024): PPO constrains updates via a clipping-based surrogate objective, while GRPO enhances long-horizon reasoning through group-based rewards.

Despite the remarkable success of RL for LLMs, ensuring stability and efficiency in optimization remains a major challenge (Yu et al., 2025; Cui et al., 2025). Recent studies have sought to better understand the underlying mechanisms of RL and proposed new methods to achieve a balance (Zheng et al., 2025; Wang et al., 2025a; Yang et al., 2025b; 2026; Ding et al., 2025a; Guo et al., 2025b; Xi et al., 2024c). For example, DAPO (Yu et al., 2025) introduces techniques such as Clip-Higher and dynamic sampling to raise the performance ceiling; Wang et al. (2025a) explore optimizing only a small subset of high-entropy tokens for improved efficiency. He et al. (2025), Cui et al. (2025), and other works (Zheng et al., 2025; Cheng et al., 2025; Liu et al., 2025) systematically investigate how to maintain entropy stability during training, thereby preserving the model's exploration ability. For off-policy RL, Roux et al. (2025b) and Arnal et al. (2025) introduce asymmetric clipping mechanisms. The most similar to our work is DCPO (Yang et al., 2025b), which adjusts token-level clipping based on token prior probabilities. However, our approach takes a holistic optimization perspective: we observe the imbalance in loss contributions and derive the Entropy-Clip Rule for the PPO objective, enabling dynamic control over global clipping bounds. We further validate the effectiveness of our method through larger-scale experiments.

## 7 CONCLUSION

In this paper, we begin by analyzing the impact of data staleness on model training through both empirical and theoretical studies. We reveal the imbalance between positive and negative samples in RL optimization, and derive as well as empirically validate the Entropy-Clip Rule for PPO-like objectives. Building on these insights, we propose BAPO, which dynamically adjusts the clipping bounds to balance positive and negative samples while preserving the model's exploratory capability during training. We conduct extensive experiments across different models and settings to validate our method. We hope our work provides key insights for the LLM RL community.

ACKNOWLEDGMENTS

The authors wish to thank the anonymous reviewers for their helpful comments. This work was partially funded by Shanghai Municipal Science and Technology Major Project 2025SHZDZX025G07, National Natural Science Foundation of China (No.62476061, 62576106, 62376061), Shanghai Rising-Star Program (23QA1400200), and Natural Science Foundation of Shanghai (23ZR1403500).

ETHICS STATEMENT

This research introduces an RL methodology designed to augment reasoning capabilities. However, we recognize that it may inadvertently strengthen other capabilities, including those with potential for malicious use. We firmly state that this work is intended for ethical and constructive purposes. Users of this method bear the full responsibility for ensuring it is applied in a safe, fair, and harmless manner. Any misuse of this method is strictly against the intent of the authors.

REPRODUCIBILITY STATEMENT

We have describe our method and the hyperparameters in §4 and §5. To support reproducibility, we will open-source our code. The datasets used for RL experiments are already publicly available.

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

## A   THE USE OF LARGE LANGUAGE MODELS

LLMs are utilized in this manuscript for partial grammatical checks and language polishing. The authors are fully responsible for the final content.

## B    PERFORMANCE ON OCTOTHINKER-LLAMA

We illustrate the training dynamics on OctoThinker-Llama in Figure 13. Since Llama family models behave badly in RL training, we choose the model after mid-training (Wang et al., 2025b) to show the robustness of BAPO. We can find that BAPO provides consistent and significant improvement in training. For training details, we set the low bound as $0.8$-$0.9$, high bound as $1.2$-$2.0$, and target positive loss contribution as $0.45$.

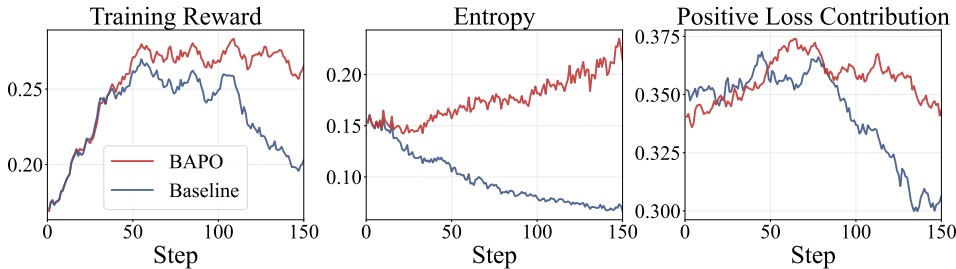

Figure 13: Training dynamics of OctoThinker-Llama-3B-Long-Zero.

## C    PERFORMANCE ON GENERAL REASONER DATASET

We illustrate the training dynamics of R1-Distill-Qwen2.5-7B trained on the General Reasoner Dataset Ma et al. (2025), which spans a wide range of domains such as physics, chemistry, finance, and electronics. Figure 14 shows that BAPO effectively stabilizes the training process and yields substantial gains across these diverse domains.

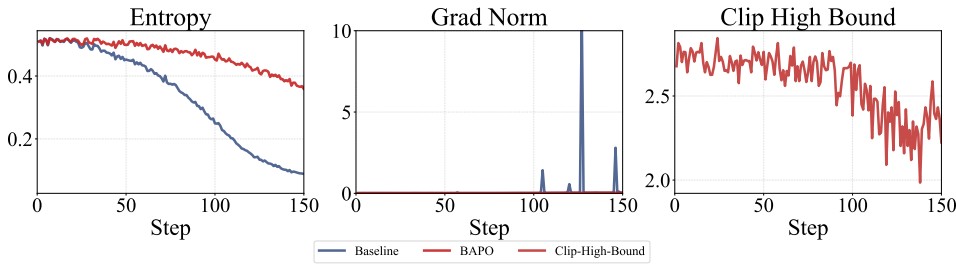

Figure 14: Training dynamics on General-Reasoner.

## D    COMPARISON WITH OTHER ALGORITHMS

Here, we present the results comparing BAPO with other algorithms.

Table 4: Performance comparison of different RL variants on the BP-Math-7B backbone. BAPO achieves consistent improvements over SFT and other state-of-the-art baselines.

| Method | AIME 24 | AIME 25 | AVG |
|---|---|---|---|
| SFT | 66.9 | 59.0 | 62.9 |
| GRPO | 69.2 | 59.2 | 64.2 |
| DAPO | 68.3 | 60.8 | 64.6 |
| TOPR | 68.8 | 60.4 | 64.6 |
| DCPO | 68.8 | 57.9 | 63.4 |
| **BAPO (Ours)** | **70.8** | **62.5** | **66.7** |

# E    ROBUSTNESS TO HYPERPARAMETERS

1. **Positive Token Ratio** $\rho_0$**:** We varied $\rho_0$ around the default value of 0.4. Results in Table 5 show minimal fluctuation in performance, suggesting the method is robust as long as a reasonable proportion of positive signal is maintained.

Table 5: Ablation studies on positive token contribution threshold $\rho_0$.

| $\rho_0$ | AIME 2024 | AIME 2025 | Average |
|---|---|---|---|
| 0.4 (default) | 70.8 | 62.5 | 66.7 |
| 0.45 | 69.6 | 63.8 | 66.7 |
| 0.50 | 71.3 | 60.8 | 66.1 |

2. **Clipping Ranges:** We tested various intervals for the lower and upper constraints. As shown in Table 6, BAPO consistently achieves high scores, proving its ability to find effective clipping points automatically.

Table 6: Ablation studies on movable range of clipping bounds.

| $[a_-, b_-]$ | $[a_+, b_+]$ | AIME 2024 | AIME 2025 | Average |
|---|---|---|---|---|
| $[0.6, 0.9]$ (default) | $[1.2, 3.0]$ (default) | 70.8 | 62.5 | 66.7 |
| $[0.5, 0.9]$ | $[1.2, 3.0]$ | 70.8 | 60.8 | 65.8 |
| $[0.6, 0.8]$ | $[1.2, 3.0]$ | 71.3 | 59.6 | 65.5 |
| $[0.6, 0.9]$ | $[1.0, 3.0]$ | 71.3 | 62.5 | 66.9 |
| $[0.6, 0.9]$ | $[1.2, +\infty]$ | 72.1 | 60.4 | 66.3 |

3. **Step Sizes** $\delta$**:** We analyzed the impact of varying the step sizes used for updating bounds. Table 7 confirms that the algorithm is insensitive to minor variations in $\delta_1$ and $\delta_2$.

Table 7: Ablation studies on step sizes $\delta_1$ and $\delta_2$.

| $\delta_1$ | $\delta_2$ | AIME 2024 | AIME 2025 | Average |
|---|---|---|---|---|
| 0.05 (default) | 0.02 (default) | 70.8 | 62.5 | 66.7 |
| 0.02 | 0.02 | 71.3 | 62.5 | 66.9 |
| 0.05 | 0.05 | 70.8 | 62.5 | 66.7 |

# F    DERIVATION FROM EQUATION 4 TO EQUATION 5

## F.1    RESTATING THE PPO OBJECTIVE (EQUATION 4)

In large-scale reinforcement learning for language models, the PPO surrogate objective is widely used:

$$J^{\text{PPO}}(\theta) = \mathbb{E}_{x \sim \mathcal{D},\, y \sim \pi_{\text{rollout}}(\cdot|x)} \left[ \sum_{t=1}^{T} \min\left(r_t \cdot A_t,\ \text{clip}(r_t, 1-\varepsilon, 1+\varepsilon) \cdot A_t\right) \right], \qquad (9)$$

where $r_t = \frac{\pi_\theta(y_t|x, y_{<t})}{\pi_{\text{rollout}}(y_t|x, y_{<t})}$, $A_t$ denotes the advantage function, and $\varepsilon \in (0,1)$ controls the clipping range.

## F.2    DECOMPOSING THE PPO LOSS STRUCTURE

Fixing a particular time step $t$, we can rewrite:

$$J^{\text{PPO}}(\theta) = \sum_{x, y_t} \pi_{\text{rollout}}(y_t \mid x, y_{<t}) \cdot \min\left(r_t \cdot A_t, \text{clip}(r_t, 1-\varepsilon, 1+\varepsilon) \cdot A_t\right). \qquad (10)$$

Note that when $r_t$ lies within the clipping interval, $\text{clip}(r_t, \cdot, \cdot) = r_t$, and thus the two terms are equal. When $r_t$ exceeds the interval, the second term becomes a constant, resulting in zero gradient with respect to $\theta$. Meanwhile, observe that $r_t \cdot \pi_{\text{rollout}} = \pi_\theta$. Therefore, we can express the objective in a piecewise form based on the sign of $A_t$:

$$
\begin{aligned}
J^{\text{PPO}}(\theta) = \sum_{x, y_t} \Big[ &\mathbb{I}\{A_t > 0\} \cdot \mathbb{I}\{r_t \leq 1 + \varepsilon\} \cdot \pi_\theta(y_t \mid x, y_{<t}) \cdot A_t \\
&+ \mathbb{I}\{A_t < 0\} \cdot \mathbb{I}\{r_t \geq 1 - \varepsilon\} \cdot \pi_\theta(y_t \mid x, y_{<t}) \cdot A_t \Big].
\end{aligned}
\tag{11}
$$

In this way, the expectation form is converted into a summation over sampled pairs $(x, y_t)$, and the behavioral policy term is eliminated, paving the way for the gradient derivation in the next section.

### F.3 TAKING THE GRADIENT WITH RESPECT TO PARAMETERS (EQUATION 5)

For the PPO objective in Equation 11, when differentiating with respect to $\theta$, only $\pi_\theta(y_t \mid x, y_{<t})$ depends on $\theta$. Thus we have:

$$
\begin{aligned}
\nabla_\theta J^{\text{PPO}} = \sum_{x, y_t} \Big[ &\mathbb{I}\{A_t > 0\} \cdot \mathbb{I}\{r_t \leq 1 + \varepsilon\} \cdot A_t \cdot \nabla_\theta \pi_\theta(y_t \mid x, y_{<t}) \\
&+ \mathbb{I}\{A_t < 0\} \cdot \mathbb{I}\{r_t \geq 1 - \varepsilon\} \cdot A_t \cdot \nabla_\theta \pi_\theta(y_t \mid x, y_{<t}) \Big].
\end{aligned}
\tag{12}
$$

Using the log-likelihood identity $\nabla_\theta \pi_\theta = \pi_\theta \nabla_\theta \log \pi_\theta$, the above becomes:

$$
\begin{aligned}
\nabla_\theta J^{\text{PPO}} = \sum_{x, y_t} \Big[ &\mathbb{I}\{A_t > 0\} \cdot \mathbb{I}\{r_t \leq 1 + \varepsilon\} \cdot \pi_\theta(y_t \mid x, y_{<t}) \cdot A_t \cdot \nabla_\theta \log \pi_\theta(y_t \mid x, y_{<t}) \\
&+ \mathbb{I}\{A_t < 0\} \cdot \mathbb{I}\{r_t \geq 1 - \varepsilon\} \cdot \pi_\theta(y_t \mid x, y_{<t}) \cdot A_t \cdot \nabla_\theta \log \pi_\theta(y_t \mid x, y_{<t}) \Big].
\end{aligned}
\tag{13}
$$

This corresponds precisely to Equation 5 in the PPO derivation: When $A_t > 0$ and $r_t < 1 + \varepsilon$, or when $A_t < 0$ and $r_t > 1 - \varepsilon$, the sample contributes an effective gradient; outside the clipping range, the gradient is suppressed to zero, enforcing smooth policy updates.

## G PROOFS OF EQUATION 6

### G.1 EXPLANATIONS FOR ALL VARIABLES AND EXPRESSIONS

All notation used in the following justification, including variables and expressions, is provided with detailed explanations in Table 8.

### G.2 PREPARATION: REWRITE THE PPO DERIVATIVES

To facilitate the justification of the propositions below, we rewrite the PPO loss function in the following form:

$$
\nabla J^{\text{PPO}} = \underbrace{\sum_{A(y_t)>0} \pi_\theta(y_t) \cdot \mathbb{I}\{r(y_t) < 1 + \varepsilon\} \cdot A(y_t) \cdot \nabla \log \pi_\theta(y_t)}_{\text{positive tokens}}
$$

$$
+ \underbrace{\sum_{A(y_t)<0} \pi_\theta(y_t) \cdot \mathbb{I}\{r(y_t) > 1 - \varepsilon\} \cdot A(y_t) \cdot \nabla \log \pi_\theta(y_t)}_{\text{negative tokens}}
$$

where

$$
\pi_\theta(y_t) = \pi_\theta(y_t | \boldsymbol{x}, \boldsymbol{y}_{<t}) \;,\quad r(y_t) = \frac{\pi_\theta(y_t \mid \boldsymbol{x}, \boldsymbol{y}_{<t})}{\pi_{\theta_{\text{rollout}}}(y_t \mid \boldsymbol{x}, \boldsymbol{y}_{<t})} \;,\quad A(y_t) = A(y_t | \boldsymbol{x}, \boldsymbol{y}_{<t}) \;.
$$

Table 8: Notation used in justification below.

| Category | Symbol | Meaning |
|---|---|---|
| Variables | $\pi_\theta$ | The policy parameterized by $\theta$ |
| | $\pi_{\theta_{rollout}}$ | The standard sampling policy |
| | $\boldsymbol{x}$ | Given prompt |
| | $\boldsymbol{y}$ | A T-token response generated by $\pi_\theta$ when given $\boldsymbol{x}$ |
| | $y_t$ | The t-th token of y |
| | $\eta$ | Learning rate |
| Expressions | $\pi_\theta(\cdot\|\boldsymbol{x}, \boldsymbol{y}_{<t})$ | Probability of generating token $\cdot$ under policy $\pi_\theta$ given input $\boldsymbol{x}$ and previous tokens $\boldsymbol{y}_{<t}$ |
| | $\pi_{\theta_{rollout}}(y_t\|\boldsymbol{x}, \boldsymbol{y}_{<t})$ | Probability of generating token $\cdot$ under standard sampling policy $\pi_{\theta_{rollout}}$ given input $\boldsymbol{x}$ and previous tokens $\boldsymbol{y}_{<t}$ |
| | $A(\cdot\|\boldsymbol{x}, \boldsymbol{y}_{<t})$ | The measurement of how much better(or worse) selecting token $\cdot$ is compared to the expected value under the current policy, given $\boldsymbol{x}$ and $\boldsymbol{y}_{<t}$ |
| | $\mathcal{H}(\cdot\|\boldsymbol{x}, \boldsymbol{y}_{<t})$ | The information entropy of policy $\cdot$ given $\boldsymbol{x}$ and $\boldsymbol{y}_{<t}$ |
| | $Cov_{y_t\sim\pi_\theta(\cdot\|\boldsymbol{x},\boldsymbol{y}_{<t})}(a(y_t), b(y_t))$ | The expected covariance of $a(y_t)$ and $b(y_t)$ over $y_t$ sampled from the policy $\pi_\theta$, given $\boldsymbol{x}$ and $\boldsymbol{y}_{<t}$ |
| | $\mathbb{I}(a = b)$ | Indicator function that equals 1 if $a = b$ and 0 otherwise |
| | $Q^{(\pi_\theta)}(\cdot, \boldsymbol{x})$ | The expected cumulative reward obtained by taking token $\cdot$ given input $\boldsymbol{x}$ and previous tokens under policy $\pi_\theta$ |
| | $V^{(\pi_\theta)}(\boldsymbol{x})$ | The expected return of the new taking token given input $\boldsymbol{x}$ and previous tokens under policy $\pi_\theta$ |
| | $z_{\boldsymbol{y},\boldsymbol{x}}$ | A quantity representing the cumulative weight of sequence $\boldsymbol{y}$ given input $\boldsymbol{x}$ under policy $\pi_\theta$, reflecting its contribution to the policy taken at the current optimization step |
| | $\nabla_{\theta_{y_t,\boldsymbol{x}}} J(\theta)$ | The gradient of the policy taken with respect to the logit parameter $\theta_{y_t,\boldsymbol{x}}$, representing how the policy $\pi_\theta$ should be adjusted for token $y_t$ given input $\boldsymbol{x}$ |

### G.3 PROOFS OF THE MAIN PROPOSITIONS

The following derivation is inspired by the proof framework in Cui et al. (2025). While the original work focuses mainly on the basic gradient formulation of naive REINFORCE to provide a heuristic explanation, our study advances this approach by deriving the gradient expression **specific to the PPO objective**. This refinement offers a specific, **intuitive yet theoretical** account of how policy entropy is intrinsically shaped by the interaction between token-level advantages and their sampling probabilities.

#### G.3.1 PRECLAIMS

Proofs of these three lemmas below are available in Cui et al. (2025).

**Lemma 1.** *Let the actor policy $\pi_\theta$ be a tabular softmax policy, the difference of information entropy given prompt x between two consecutive steps $k$ and $k + 1$ satisfies*

$$\mathcal{H}(\pi_\theta^{k+1}|\boldsymbol{x}, \boldsymbol{y}_{<t}) - \mathcal{H}(\pi_\theta^k|\boldsymbol{x}, \boldsymbol{y}_{<t}) \approx -\text{Cov}_{y_t\sim\pi_\theta^k(\cdot|\boldsymbol{x},\boldsymbol{y}_{<t})}\left(\log \pi_\theta^k(y_t),\ z_{\boldsymbol{y},\boldsymbol{x}}^{k+1} - z_{\boldsymbol{y},\boldsymbol{x}}^k\right).$$

**Lemma 2** (Derivative of softmax function).

$$\frac{\partial \log \pi_\theta(y_t)}{\partial \theta_{y'_t, \boldsymbol{x}}} = \mathbb{I}\{y_t = y'_t\} - \pi_\theta(y'_t)$$

**Lemma 3** (Expectation of Advantage function given prompt $x$).

$$
\begin{aligned}
\mathbb{E}_{y_t \sim \pi_\theta(\cdot|x, \boldsymbol{y}_{<t})}\left[A^{\pi_\theta}(y_t)\right] &= \mathbb{E}_{y_t \sim \pi_\theta(\cdot|x, \boldsymbol{y}_{<t})}\left[Q^{\pi_\theta}(y_t, \boldsymbol{x}) - V^{\pi_\theta}(\boldsymbol{x})\right] \\
&= \mathbb{E}_{y_t \sim \pi_\theta(\cdot|x, \boldsymbol{y}_{<t})}\left[Q(y_t, \boldsymbol{x})\right] - \mathbb{E}_{y_t \sim \pi_\theta(\cdot|x, \boldsymbol{y}_{<t})}\left[V(\boldsymbol{x})\right] \\
&= V(\boldsymbol{x}) - V(\boldsymbol{x}) \\
&= 0
\end{aligned}
$$

### G.3.2 PRINCIPLE PROPOSITIONS

**Proposition 1:** Assume the actor policy $\pi_\theta$ follows a tabular softmax policy and is optimized via the PPO objective, the difference of $z_{\boldsymbol{y}, \boldsymbol{x}}$ between two consecutive steps k and k+1 satisfies

$$z_{\boldsymbol{y}, \boldsymbol{x}}^{k+1} - z_{\boldsymbol{y}, \boldsymbol{x}}^k = \eta \cdot \pi_\theta(y_t) \cdot [A(y_t) \cdot \mathcal{X}(y_t) + C],$$

where

$$
\mathcal{X}(y_t) = \begin{cases} 1, & \textbf{\textit{if}} \ A(y_t) > 0 \ \& \ r(y_t) < 1 + \epsilon \\ & \textbf{\textit{or}} \ A(y_t) < 0 \ \& \ r(y_t) > 1 - \epsilon \\ 0, & \text{otherwise} \end{cases}
$$

and $C$ includes all clauses irrelevant to $y_t$.

**It is worth noting that $\mathcal{X}(y_t) = 0$ if and only if $y_t$ is clipped.**

*Proof.* In tabular softmax policy, each trajectory-prompt pair $(\boldsymbol{y}, \boldsymbol{x})$ is associated with an individual logit parameter $z_{\boldsymbol{y}, \boldsymbol{x}} = \theta_{y_t, \boldsymbol{x}}$. Through gradient backtracking, $z_{\boldsymbol{y}, \boldsymbol{x}}$ is updated via $z_{\boldsymbol{y}, \boldsymbol{x}}^{k+1} = z_{\boldsymbol{y}, \boldsymbol{x}}^k + \eta \cdot \nabla_{\theta_{y_t, \boldsymbol{x}}} J(\theta)$. According to the loss function of PPO, we have

$$
\begin{aligned}
z_{\boldsymbol{y}, \boldsymbol{x}}^{k+1} - z_{\boldsymbol{y}, \boldsymbol{x}}^k &= \eta \cdot \nabla_{\theta_{y_t, \boldsymbol{x}}} J_{PPO}(\theta) \\
&= \eta \cdot \mathbb{E}_{\substack{y'_t \sim \pi_\theta(\cdot|\boldsymbol{x}, \boldsymbol{y}_{<t}) \\ A(y'_t) > 0}} \left[\mathbb{I}\{r(y'_t) < 1 + \varepsilon\} \cdot \nabla_{\theta_{y_t, \boldsymbol{x}}} \log \pi_\theta(y'_t) \cdot A(y'_t)\right] \\
&\quad + \eta \cdot \mathbb{E}_{\substack{y'_t \sim \pi_\theta(\cdot|\boldsymbol{x}, \boldsymbol{y}_{<t}) \\ A(y'_t) < 0}} \left[\mathbb{I}\{r(y'_t) > 1 - \varepsilon\} \cdot \nabla_{\theta_{y_t, \boldsymbol{x}}} \log \pi_\theta(y'_t) \cdot A(y'_t)\right] \\
&= \underbrace{\eta \cdot \mathbb{E}_{y'_t \sim \pi_\theta(\cdot|\boldsymbol{x}, \boldsymbol{y}_{<t})} \left[\nabla_{\theta_{y_t, \boldsymbol{x}}} \log \pi_\theta(y'_t) \cdot A(y'_t)\right]}_{①} \\
&\quad - \underbrace{\eta \cdot \mathbb{E}_{\substack{y'_t \sim \pi_\theta(\cdot|\boldsymbol{x}, \boldsymbol{y}_{<t}) \\ A(y'_t) > 0}} \left[\mathbb{I}\{r(y'_t) > 1 + \varepsilon\} \cdot \nabla_{\theta_{y_t, \boldsymbol{x}}} \log \pi_\theta(y'_t) \cdot A(y'_t)\right]}_{②} \\
&\quad - \underbrace{\eta \cdot \mathbb{E}_{\substack{y'_t \sim \pi_\theta(\cdot|\boldsymbol{x}, \boldsymbol{y}_{<t}) \\ A(y'_t) < 0}} \left[\mathbb{I}\{r(y'_t) < 1 - \varepsilon\} \cdot \nabla_{\theta_{y_t, \boldsymbol{x}}} \log \pi_\theta(y'_t) \cdot A(y'_t)\right]}_{③} \\
&= ① - (② + ③)
\end{aligned}
\tag{8}
$$

We first perform the derivation on the term marked as ①:

$$\textcircled{1} = \eta \cdot \mathbb{E}_{y'_t \sim \pi_\theta(\cdot|\boldsymbol{x}, \boldsymbol{y}_{<t})} \left[ \frac{\partial \log \pi_\theta(y'_t)}{\partial \theta_{y_t, \boldsymbol{x}}} \cdot A(y'_t) \right]$$

$$\overset{\text{Lemma 2}}{=} \eta \cdot \sum_{y'_t} \left[ \pi_\theta(y'_t) \cdot (\mathbb{I}\{y'_t = y_t\} - \pi_\theta(y_t)) \cdot A(y'_t) \right]$$

$$= \eta \cdot \pi_\theta(y_t) \cdot \left[ (1 - \pi_\theta(y_t)) \cdot A(y_t) - \sum_{y'_t \neq y_t} \pi_\theta(y'_t) \cdot A(y'_t) \right]$$

$$= \eta \cdot \pi_\theta(y_t) \cdot \left[ A(y_t) - \sum_{y'_t} \pi_\theta(y'_t) \cdot A(y'_t) \right]$$

$$\overset{\text{Lemma 3}}{=} \eta \cdot \pi_\theta(y_t) \cdot [A(y_t) - 0]$$

$$= \eta \cdot \pi_\theta(y_t) \cdot A(y_t)$$

To keep the presentation concise, we provide only the resulting derivations of Term $\textcircled{2}$ and $\textcircled{3}$, as the detailed steps follow similarly to those for Term $\textcircled{1}$.

$$\textcircled{2} + \textcircled{3} = \eta \cdot \pi_\theta(y_t) \cdot A(y_t) \cdot (1 - \mathcal{X}(y_t))$$

$$- \eta \cdot \pi_\theta(y_t) \cdot \sum_{A(y'_t)>0} \left[ \mathbb{I}\{r(y'_t) > 1 + \varepsilon\} \cdot \pi_\theta(y'_t) \cdot A(y'_t) \right]$$

$$- \eta \cdot \pi_\theta(y_t) \cdot \sum_{A(y'_t)<0} \left[ \mathbb{I}\{r(y'_t) < 1 - \varepsilon\} \cdot \pi_\theta(y'_t) \cdot A(y'_t) \right]$$

By substituting the results of the above derivation into Clause (8), we observe that:

$$(8) = \textcircled{1} - (\textcircled{2} + \textcircled{3})$$

$$= \eta \cdot \pi_\theta(y_t) \cdot \Big\{ A(y_t) \cdot \mathcal{X}(y_t)$$

$$+ \sum_{A(y'_t)>0} \left[ \mathbb{I}\{r(y'_t) > 1 + \varepsilon\} \cdot \pi_\theta(y'_t) \cdot A(y'_t) \right]$$

$$+ \sum_{A(y'_t)<0} \left[ \mathbb{I}\{r(y'_t) < 1 - \varepsilon\} \cdot \pi_\theta(y'_t) \cdot A(y'_t) \right] \Big\}$$

By grouping all elements unrelated to $y_t$ into $C$, we are able to successfully establish our proposition.

$\square$

Building on Proposition 1, we establish the relationship between policy entropy and the covariance of specific tokens, which is stated as Proposition 2 below.

**Proposition 2 (Equation 6):** Let the actor policy $\pi_\theta$ be tabular softmax policy, and $\pi_\theta$ is updated via PPO objective, the difference of information entropy given prompt $x$ and trajectory part $y_{<t}$ between two consecutive steps k and k+1 satisfies

$$\mathcal{H}(\pi_\theta^{k+1}|\boldsymbol{x}, \boldsymbol{y}_{<t}) - \mathcal{H}(\pi_\theta^k|\boldsymbol{x}, \boldsymbol{y}_{<t}) \approx -\eta \cdot \text{Cov}_{y_t \sim \pi_\theta^k(\cdot|\boldsymbol{x},\boldsymbol{y}_{<t})} \left( \log \pi_\theta^k(y_t), A(y_t) \cdot \mathcal{X}(y_t) + C \right).$$

*Proof.* Leveraging the conclusions of Lemma 1 and Proposition 1, we find that, under policy optimization and iteration via the PPO algorithm, the following relationship is satisfied:

$$z_{\boldsymbol{y},\boldsymbol{x}}^{k+1} - z_{\boldsymbol{y},\boldsymbol{x}}^k = \eta \cdot (A(y_t) \cdot \mathcal{X}(y_t) + C).$$

Applying this into Lemma 1, we have

$$\mathcal{H}(\pi_\theta^{k+1}|\boldsymbol{x}, \boldsymbol{y}_{<t}) - \mathcal{H}(\pi_\theta^k|\boldsymbol{x}, \boldsymbol{y}_{<t}) \approx -\eta \cdot \mathrm{Cov}_{y_t \sim \pi_\theta^k(\cdot|\boldsymbol{x}, \boldsymbol{y}_{<t})} \left( \log \pi_\theta^k(y_t), A(y_t) \cdot \mathcal{X}(y_t) + C \right).$$

$\square$

### G.4 ANALYSIS

#### G.4.1 DIRECT ANALYSIS: WHY VARYING $\varepsilon$ ALTERS ENTROPY?

We begin by examining the covariance of the clipped token, denoted as $\alpha$.

Based on the observation stated above, the contribution of $\alpha$ to the entropy can be expressed as:

$$-\eta \cdot \pi_\theta^k(\alpha) \cdot \mathrm{Cov}\big(\log \pi_\theta^k(\alpha), C\big) = 0,$$

which indicates that only the retained tokens contribute to the overall entropy.

In other words, we manipulate the number of tokens that can contribute to the entropy by altering the parameter $\varepsilon$.

#### G.4.2 ADVANCED ANALYSIS : WHICH TYPE OF TOKENS MATTER MOST FOR ENTROPY?

To understand how individual tokens contribute to the overall entropy, we first revisit the Proposition G.3.2 established above. In this section, we provide a more precise definition of tokens with low/high probabilities and advantages. It should be noted that in the analysis experiment (Figure 5), we adopt the naive REINFORCE algorithm without clipping. Consequently, tokens with high or low advantages are defined according to the sign of their advantage values, i.e., $> 0$ for high advantage and $< 0$ for low advantage.

$$\mathcal{H}(\pi_\theta^{k+1}|\boldsymbol{x}, \boldsymbol{y}_{<t}) - \mathcal{H}(\pi_\theta^k|\boldsymbol{x}, \boldsymbol{y}_{<t}) \approx -\eta \cdot \mathrm{Cov}_{y_t \sim \pi_\theta^k(\cdot|\boldsymbol{x}, \boldsymbol{y}_{<t})} \left( \log \pi_\theta^k(y_t), A(y_t) \cdot \mathcal{X}(y_t) + C \right)$$

$$= -\eta \cdot \sum_{p=1}^T \pi_\theta^k(y_p|\boldsymbol{x}, \boldsymbol{y}_{<t}) \cdot \big( \log \pi_\theta^k(y_p) - \mathbb{E}_{y_i \sim \pi_\theta^k(\cdot|\boldsymbol{x}, \boldsymbol{y}_{<t})}[\log \pi_\theta^k(y_i)]\big)$$

$$\cdot \big( A(y_p) \cdot \mathcal{X}(y_p) - \mathbb{E}_{y_i \sim \pi_\theta^k(\cdot|\boldsymbol{x}, \boldsymbol{y}_{<t})}[A(y_i) \cdot \mathcal{X}(y_i)]\big).$$

where T is the size of the dictionary.

For convenience, we denote $\mathbb{E}_{y_i}$ as $\mathbb{E}_{y_i \sim \pi_\theta^k(\cdot|\boldsymbol{x}, \boldsymbol{y}_{<t})}$. As only retained tokens contribute to the entropy, we focus only on tokens that are not clipped. We begin by making the following simplification:

$$\mathbb{E}_{y_i}(A(y_i) \cdot \mathcal{X}(y_i)) = \mathbb{E}_{y_{\mathrm{clipped}}}(A(y_i) \cdot 0) + \mathbb{E}_{y_{\mathrm{retained}}}(A(y_i) \cdot 1) = \mathbb{E}_{y_{\mathrm{retained}}}(A(y_i)).$$

So for a selected token $y_s$, its contribution to the overall entropy can be expressed as:

$$-\eta \cdot \pi_\theta(y_s) \cdot (\log \pi_\theta(y_s) - \mathbb{E}_{y_i}(\log \pi_\theta(y_i))) \cdot (A(y_s) - \mathbb{E}_{y_{\mathrm{retained}}} A(y_{\mathrm{retained}})).$$

Next, we analyze how different types of tokens contribute to the overall entropy. To avoid ambiguity, we first give strict definitions that distinguish between tokens with high/low probabilities and tokens with high/low advantages.

**Definition 1.** *For a token $y_s$, we classify it as follows:*

- ***High advantage:*** *if*
$$A(y_s) > \mathbb{E}_{y_{\mathrm{retained}}} A(y_{\mathrm{retained}})$$
*Otherwise, it is called* low advantage.

- **High probability:** *if*
$$\pi_\theta(y_s) > \exp(\mathbb{E}_{y_i}(\log \pi_\theta(y_i))))$$
*Otherwise, it is called* low probability.

Secondly, we present two propositions that directly follow from the above definitions.

**Proposition 3.** For a token $y_s$, we have

$$A(y_s) - \mathbb{E}_{y_{\text{retained}}} A(y_{\text{retained}}) \begin{cases} > 0, & \text{if } y_s \text{ is a high-advantage token,} \\ < 0, & \text{if } y_s \text{ is a low-advantage token.} \end{cases}$$

**Proposition 4.** For a token $y_s$, we have

$$\pi_\theta(y_s) \cdot (\log \pi_\theta(y_s) - \mathbb{E}_{y_i}(\log \pi_\theta(y_i))) \begin{cases} > 0, & \text{if } y_s \text{ is a high-probability token,} \\ < 0, & \text{if } y_s \text{ is a low-probability token.} \end{cases}$$

*Proof.* Let us denote

$$C = \mathbb{E}_{y_i}(\log \pi_\theta(y_i))),$$

which is independent of $y_s$, and let $x = \pi_\theta(y_s)$. As $\pi_\theta(y) < 1$ for every $y$, $C < 0$. Consider the function

$$f(x) = x \cdot (\log(x) - C).$$

Figure 15 illustrates the behavior of this function.

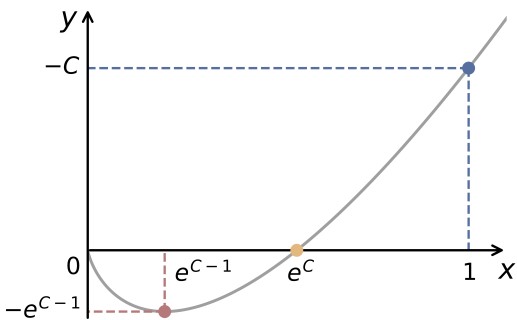

Figure 15: Graph of the function $f(x) = x(\log x - C)$.

The proposition follows directly from the properties of $f(x)$ as observed in the figure. □

Due to the propositions given above, we have the table below:

$$\Delta \mathcal{H}(y_s) \approx -\eta \cdot \underbrace{\pi_\theta(y_s) \cdot (\log \pi_\theta(y_s) - \mathbb{E}_{y_i}(\log \pi_\theta(y_i)))}_{\text{④}} \cdot \underbrace{(A(y_s) - \mathbb{E}_{y_{\text{retained}}} A(y_{\text{retained}}))}_{\text{⑤}}$$

Table 9: Influence of token characteristics on $\Delta \mathcal{H}(y_s)$. The "prob" denotes the probability $\pi_\theta(y_s)$, and the "adv" represents the advantage $A(y_s)$.

| Token properties | ④ | ⑤ | $\Delta \mathcal{H}(y_s)$ $(-\eta \cdot ④ \cdot ⑤)$ |
|---|---|---|---|
| high prob, high adv | $> 0$ | $> 0$ | $< 0$ |
| high prob, low adv | $> 0$ | $< 0$ | $> 0$ |
| low prob, high adv | $< 0$ | $> 0$ | $> 0$ |
| low prob, low adv | $< 0$ | $< 0$ | $< 0$ |

It should be noted that a token $y_s$ **decreases** the entropy if $\Delta\mathcal{H}(y_s) < 0$, and **increases** it otherwise.

Therefore, we observe that tokens which are positive with high probabilities and high advantages, or negative with low probabilities and low advantages, contribute to a reduction in the overall entropy. Conversely, positive tokens with high probabilities but low advantages, and negative tokens with high probabilities but low advantages, contribute to an increase in the overall entropy. This observation justifies the statement made in the main part of the thesis.

