# OpenReview forum: "BAPO: Stabilizing Off-Policy Reinforcement Learning for LLMs via Balanced Policy Optimization with Adaptive Clipping"
_ICLR.cc/2026/Conference — ICLR 2026 Poster_

### Official Review · Reviewer_Zvwr · 2025-10-23

**Soundness:** 3
**Presentation:** 3
**Contribution:** 3
**Rating:** 6
**Confidence:** 3

**Summary:**

The paper proposes a new adaptive and asymmetric clipping for PPO algorithm applied to LLM. Indeed the authors first observe that a common symmetrical clipping $clip(r_t, 1-\epsilon,1+\epsilon)$ for some $\epsilon > 0$ implies a decrease of policy's entropy with its eventual collapse, which prevents from further exploration. They argue that it happens because of misbalance between positive and negative tokens. The final algorithm BAPO adaptively tunes the clipping bounds to include more low probability positive tokens, and potentially reducing the number of negative tokens that are retained for further update steps. This approach is theoretically motivated and is further validated on AIME 2024 and AIME 2025 benchmarks, showing the consistent improvement over GRPO and SFT, demonstrating even some competitive performance compared to some proprietary models.

Overall, the paper is easy to read, providing new interesting results, but it can benefit from more comparisons with other prior assymetric clipping mechanisms. Therefore, I recommend weak accept.

**Strengths:**

The paper is nicely written and in a concise manner. BAPO represents an easy to implement and flexible clipping mechanism for finetuning LLM with RL. It is well motivated, with some theoretical insights, and its advantage over a manual symmetrical clipping is evident. Their models trained with BAPO achieve on-par results on AIME 2024 and 2025 compared to some proprietary algorithms.

**Weaknesses:**

Despite that the paper mentions other works that consider asymmetric clipping mechanisms, like TOPR, DAPO, and even dynamic clipping — DRPO, there is no systematic comparison between different methods for fixed original models like BP-Math-7B or BP-Math-32B. The only baselines that the authors are comparing to are GRPO, SFT and a single fixed asymmetric clipping with strictly predefined bounds.

Minor:

- L141: provide the specific function for A_t that you use
- Figure11: why not checking [0.8, 1.5] as in figure 7?
- L841: provide a more precise definition for $z_{y,x}$

**Questions:**

- Is there a computational overhead of computing BAPO? How big?
- How does BAPO perform for longer training, does it only delay the entropy collapse or the entropy is expected to stabilise at some small non-zero value?

---

> ### Author Response · Authors · 2025-11-24
> **Response to Reviewer Zvwr. Part [1/2]**
>
> We appreciate your valuable time, feedback, and for highlighting our strengths (e.g., the insightful motivation, the flexible method BAPO, and the strong performance). In the following, we will carefully respond to your questions.
>
> ### **# Question about more baselines**
>
> We appreciate your constructive suggestions; they are very helpful to us. Following your suggestion, we conducted a systematic comparison of different methods (TOPR [1], DAPO [2], and DCPO [3]) using a fixed base model (BP-Math-7B). The results are shown in the table below. We find that our method outperforms these baselines and yields more stable training, highlighting the advantages of our approach. We will update the manuscript accordingly.
>
> |             |  AIME24  |  AIME25  |    AVG   |
> |:-----------:|:--------:|:--------:|:--------:|
> |     SFT     |   66.9   |    59    |   62.9   |
> |     GRPO    |   69.2   |   59.2   |   64.2   |
> |     DAPO    |   68.3   |   60.8   |   64.6   |
> |     TOPR    |   68.8   |   60.4   |   64.6   |
> |     DCPO    |   68.8   |   57.9   |   63.4   |
> | BAPO (Ours) | **70.8** | **62.5** | **66.7** |
>
> ---
>
> ### **# Question about checking [0.8, 1.5] in Figure 11**
>
> We appreciate your valuable suggestion, which made us realize this omission. We have added the results to Figure 11 in the latest version of our manuscript, and we also present them in the table below. We find that this method achieves decent improvements, but it still falls short of our approach.
>
> | Staleness |       Method       | AIME24 | AIME25 |
> |:---------:|:------------------:|:------:|:------:|
> |     2     | Clip w/ [0.8, 1.5] |  58.8  |  40.0  |
> |           |     BAPO (ours)    |  60.9  |  44.2  |
> |     4     | Clip w/ [0.8, 1.5] |  58.9  |  40.8  |
> |           |     BAPO (ours)    |   62   |  43.3  |
>
> ---
>
> ### **# Question about computational overhead of adjusting clipping bounds**
>
> Thank you for your inquiry regarding the computational overhead.  To validate this in practice, we measured the time spent dynamically computing the clipping bounds as a fraction of the total time per step.
>
> | Response Length | Time of Adjusting Clipping Bounds | Total Time of the Step | Proportion |
> |-----------------|-----------------------------------|------------------------|------------|
> | 32k             | 7.2s                              | 674.5s                 | 1.07%      |
> | 8k              | 2.1s                              | 156.8s                 | 1.34%      |
>
> According to the table above, we can clearly observe that the time spent adjusting the clipping bounds accounts for only a very small portion of the total time per step. Therefore, the additional computational overhead is minimal and can be regarded as nearly negligible.
>
> ---
>
> ### **# Question about how does BAPO perform for longer training**
>
> Thank you very much for your question—it helped us realize that our description was not fully clear. In fact, BAPO enables effective positive learning by balancing positive and negative examples, rather than merely compressing the vast language sampling space. In addition, by adjusting the tokens involved in training with different probabilities, we stabilize the entropy, allowing the model to retain its exploration ability. These two effects work together to achieve stable training. In practice, we rarely observe BAPO collapsing during training. We typically stop the training manually because it has run for many steps or too long, rather than because it collapses. We also conducted thorough hyperparameter analyses for BAPO (also shown in the **Point #4 of the General Response**), demonstrating its broad applicability and wide usability.
>
> (1) Ablation on positive token contribution threshold $\rho_0$:
>
> |      $\boldsymbol{\rho_0}$     | AIME24 | AIME25 | Average |
> |:-----------:|:------:|:------:|:-------:|
> | 0.4 (paper) |  70.8  |  62.5  |   66.7  |
> |     0.45    |  69.6  |  63.8  |   66.7  |
> |     0.5     |  71.3  |  60.8  |   66.1  |
>
> (2) Ablation on movable range of clipping bounds $[a_-, b_-]$ and $[a_+, b_+]$:
>
> |      $\boldsymbol{[a_-, b_-]}$      |      $\boldsymbol{[a_+, b_+]}$      | AIME24 | AIME25 | Average |
> |:------------------:|:------------------:|:------:|:------:|:-------:|
> | [0.6, 0.9] (paper) | [1.2, 3.0] (paper) |  70.8  |  62.5  |   66.7  |
> |     [0.5, 0.9]     |     [1.2, 3.0]     |  70.8  |  60.8  |   65.8  |
> |     [0.6, 0.8]     |     [1.2, 3.0]     |  71.3  |  59.6  |   65.5  |
> |     [0.6, 0.9]     |     [1.0, 3.0]     |  71.3  |  62.5  |   66.9  |
> |     [0.6, 0.9]     |     [1.2, +inf]    |  72.1  |  60.4  |   66.3  |
>
> (3) Ablation on step sizes $\delta_1$ and $\delta_2$:
>
> |    $\boldsymbol{\delta_1}$    |    $\boldsymbol{\delta_2}$    | AIME24 | AIME25 | Average |
> |:------------:|:------------:|:------:|:------:|:-------:|
> | 0.05 (paper) | 0.02 (paper) |  70.8  |  62.5  |   66.7  |
> |     0.02     |     0.02     |  71.3  |  62.5  |   66.9  |
> |     0.05     |     0.05     |  70.8  |  62.5  |   66.7  |

---

> ### Author Response · Authors · 2025-11-24
> **Response to Reviewer Zvwr. Part [2/2]**
>
> ### **# Question about the Definition of Advantage $A_t$**
>
> Thank you for this insightful question. We clarify the definition of the advantage $A_t$ from both theoretical and practical perspectives below:
>
> 1. Theoretical Definition (Standard $Q-V$ Formulation)
>
>     In our theoretical analysis and derivations (e.g., Eq. 2 and Lemma 1), we adhere to the standard Reinforcement Learning definition where the advantage $A_t$ represents the relative value of an action compared to the baseline:
>
>     $$A_t(x, y_t) = Q^{\pi_\theta}(x, y_{t}) - V^{\pi_\theta}(x).$$
>
>     This standard formulation ensures that all theoretical properties and gradient derivations discussed in our paper hold true under general RL frameworks.
>
> 2. Practical Implementation (GRPO)
>
>     For our experiments, we adopt Group Relative Policy Optimization (GRPO) to estimate this advantage. GRPO is a widely used and highly successful method in the field of LLM reasoning [4-5], designed to reduce variance and stabilize training without requiring a separate value network.
>
>     Specifically, GRPO samples a group of $K$ responses $\{y_1, \dots, y_K\}$ for each prompt and estimates the advantage by normalizing the rewards within the group. The advantage $A_t$ for the $t$-th token in sequence $y_i$ is calculated as:
>
>     $$A_t = \frac{r(y_i) - \text{mean}(\{r(y_1), \dots, r(y_K)\})}{\text{std}(\{r(y_1), \dots, r(y_K)\})}.$$
>
>     This estimation serves as a robust proxy for $Q-V$ in practice, effectively handling the sparse reward nature of reasoning tasks and contributing to the significant performance gains observed in our results.
>
> ---
>
> ### **# Question about a more precise definition for $z_{y,x}$**
>
> 1. Mathematical Definition (Logit Parameter)
>
>     In the context of our theoretical analysis (Lemma 1 and its Proof), we adopt the Tabular Softmax Policy assumption. Here, $z_{\boldsymbol{y}, \boldsymbol{x}}$ is defined as the learnable logit parameter $\theta$ corresponding to the specific action (token) $y_t$ given the context $(\boldsymbol{x}, \boldsymbol{y}_{<t})$.
>
>     Mathematically, the policy is parameterized as:
>
>     $$\pi_\theta(y_t | x, y_{<t}) = \frac{\exp (z_{y, x})}{\sum_{y'} \exp(z_{y', x})},$$
>
>     where $z_{\boldsymbol{y}, \boldsymbol{x}} \equiv \theta_{y_t, (\boldsymbol{x}, \boldsymbol{y}_{<t})}$.
>
> 2. Dynamic Definition (Accumulated Gradient)
>
>     From the perspective of optimization dynamics (as detailed in the Proof of Lemma 1), $z_{\boldsymbol{y}, \boldsymbol{x}}$ represents the cumulative state of the policy parameters. Its evolution between training steps $k$ and $k+1$ is determined by the gradient update rule:
>
>     $$z_{\boldsymbol{y}, \boldsymbol{x}}^{k+1} = z_{\boldsymbol{y}, \boldsymbol{x}}^{k} + \eta \cdot \nabla_{\theta} J(\theta).$$
>
>     Therefore, $z_{\boldsymbol{y}, \boldsymbol{x}}$ can be interpreted as the accumulator of gradient updates (scaled by the learning rate $\eta$) applied to the specific logit associated with the trajectory-prompt pair. This variable allows us to track how the policy's raw preference for a token shifts before the softmax normalization is applied.
>
> ---
>
> We once again thank you for your valuable comments! If you have any further questions, please feel free to let us know and we will do our utmost to respond. If you find our replies satisfactory, we kindly ask that you consider updating your score and confidence accordingly.
>
> ---
>
> **References**
>
> [1] Trust Region Policy Optimization.
>
> [2] DAPO: An Open-Source LLM Reinforcement Learning System at Scale.
>
> [3] DCPO: Dynamic Clipping Policy Optimization.
>
> [4] DeepSeekMath: Pushing the Limits of Mathematical Reasoning in Open Language Models.
>
> [5] DeepSeek-R1: Incentivizing Reasoning Capability in LLMs via Reinforcement Learning.

---

> ### Author Response · Authors · 2025-11-27
> **Invitation to discussion**
>
> Dear Reviewer Zvwr,
>
> We would like to thank you for the valuable and constructive suggestions, and appreciate that you agree on the strengths of our paper. During the rebuttal, we performed additional experiments and provided detailed clarifications to address your concerns. The modifications in our response are summarized as follows:
>
> - We conducted a systematic comparison of different methods (TOPR, DAPO, and DCPO) using a fixed base model (BP-Math-7B).
>
> - We added experiments using Clip w/ [0.8, 1.5] under the setting of Figure 11.
>
> - We measured the time spent dynamically computing the clipping bounds to demonstrate that the additional computational overhead is minimal and can be regarded as nearly negligible.
>
> - We added clarifications on how BAPO performs for longer training, focusing on its stability.
>
> - We added detailed definitions of advantage $A_t$ and $z_{y, x}$ in response to your questions.
>
> We are wondering if the responses have addressed your concerns and would appreciate it if you consider raising your final rating. Looking forward to your further feedback!
>
> Best Regards,
>
> Authors of ICLR 2026 Conference Paper 20329

---

### Official Review · Reviewer_drrk · 2025-10-24

**Soundness:** 2
**Presentation:** 4
**Contribution:** 2
**Rating:** 4
**Confidence:** 4

**Summary:**

The paper tackles instability in RL for LLMs with respect to a fundamental challenge in off-policy training: data staleness or distribution shift, i.e., the distribution of rollout policy differs from the distribution of the optimization target policy. The paper proposes that the instability mainly comes from two sources: i) imbalanced positive and negative samples, and ii) lack of exploration. To address both issues from empirical practice, the authors propose BAPO, which dynamically adjusts lower/upper importance sampling weight clip bounds to balance the negative and positive tokens. Experiments on reasoning tasks show competitive improvement over open baselines without clip-bound adjustment.

**Strengths:**

1. The empirical performance with an asymmetric clip constant is promising.
2. Some intuition-level justification from the policy gradient contribution perspective and the entropy dynamic perspective is given to pave the motivation.
3. The paper is quite well-written and easy to follow.

**Weaknesses:**

I'm somewhat skeptical of both perspectives on the training instability discussed in the paper:

1. From the policy gradient perspective, e.g., Eq. (3), in general RL, both positive and negative samples are important in estimating the value function gradient. Taking out either part will result in biased estimation and harm the performance improvement. On the other hand, if a random minibatch contains only negative samples, the policy can still be improved in expectation. So, the dominance of negative samples does not necessarily prevent learning and lead to training instability or failure.

2. The decrease of entropy also does not directly imply instability of training. The entropy decreases as the policy converges to a deterministic policy, which may or may not be the optimal solution. Therefore, entropy collapse should be expected for all RL training. If the policy converges to the optimal policy, the entropy also collapses, but we should not witness instability or sub-optimality. The instability could come from other places, such as the distribution shift or simply the high variance of using importance sampling. Whether to preserve or decrease entropy is the classic trade-off in exploration and exploitation, which this paper oversimplifies.

3. Essentially, the problem studied in this paper is somewhat artificial and stems from the clip-trick instead of the RL problem itself. In the ideal case, one should clip outlier events symmetrically over the probability measure such that the clipped estimation is still an unbiased estimator of the genuine policy gradient, but symmetric clipping on numerical values fails to coincide, since the irregular distribution of advantages in practice, and potentially depends on the state as well. These shortcomings have been quite well-studied in classic winsorization literature with application to RL, and, quite surprisingly to me, the paper does not discuss any of them. Therefore, I do not see much ideological contribution in the field. See below:


4. It seems all the downsides and limitations of PPO-clip mentioned in the paper, i.e., one single trust region for all states, lack of exploration due to fast entropy decrease, could be naturally solved with PPO-KL or TRPO, i.e., posing the trust region on policy instead of the action of each state, and using the trust region to prevent fast entropy decrease. It is very unclear why the authors do not discuss this approach. The paper should include a comprehensive discussion (from theory to empirical evaluation) between dynamic clipping and the policy trust region approaches, to convince the value of their proposed approach.

Orenstein, Robust Importance Sampling with Adaptive Winsorization, 2018

Vehtari et al., Pareto Smoothed Importance Sampling, 2015

Su et al., Doubly robust off-policy evaluation with shrinkage, 2020

**Questions:**

See Weaknesses.

Some minor question:

1. From Figure 8, the fluctuation of both bounds across time is very minor compared to the asymmetry between higher and lower bounds. It seems we could also use a stationary asymmetric clip constant across time. Why would practitioners resort to dynamic clipping, which needs to compute the clip constant every step, versus a stationary asymmetric clip? Why are the authors in favor of the first approach?

2. In (8), why do you use $c_{low} = 0$ in the numerator?

---

> ### Author Response · Authors · 2025-11-24
> **Response to Reviewer drrk. Part [1/3]**
>
> We appreciate your valuable time, insights, and for highlighting our strengths. In the following, we will carefully respond to your questions.
>
> ### **# Question about positive and negative samples in policy gradient**
>
> Thank you for your question from the policy-gradient perspective. We agree that both positive and negative samples are important for estimating the gradient of the value function, especially in general RL.
>
> However, we would like to clarify that our work is situated in the specific context of RL for LLMs, where RL optimization typically faces many challenges. Prior work has pointed out that in LLM RL, signals from positive samples are often more important than those from negative samples because they stabilize training [7][18]. Intuitively, negative reward signals tend to shrink the sampling space so that the model can more easily find positive samples. However, the sampling space of LLMs is usually extremely large (due to a large vocabulary and long reasoning trajectories). With only sparse final reward signals, batches consisting solely of negative samples provide a relatively weak learning signal, which slows convergence and causes instability.
>
> Therefore, our method BAPO actively maintains a proportion of positive samples to achieve a balance, because they are practically crucial for stable and efficient learning. Positive samples provide a strong, directional signal that anchors the policy toward known high-reward regions, preventing collapse and guiding exploration more effectively.
>
> To further support this, we conducted comparative experiments on the DeepSeek-R1-Distill-7B model with a maximum sequence length of 8k.
>
> |             Method             | AIME24 | AIME25 |  AVG |
> |:------------------------------:|:------:|:------:|:----:|
> | Learning from positive samples |  49.6  |  29.5  | 39.5 |
> | Learning from negative samples |  42.5  |  27.2  | 34.9 |
> |           Ours (BAPO)          |  51.3  |  34.6  | 43.0 |
>
> As shown in the table, learning directly from positive samples is more efficient than learning from negative ones. This finding is also supported by prior RL work [1][7][19]. **Moreover, we show that our balanced positive–negative optimization strategy performs better than optimizing only one side, which further validates our view that both signals matter—particularly for maintaining exploration capability in the later stages of training.**
>
> ---
>
> ### **# Question about policy trust region approaches**
>
> Thank you for your suggestion; it is very helpful to us. We conducted several experiments on DeepSeek-R1-Distill-Qwen-7B model, and found that this strategy does improve training stability to some extent. However, the final model performance remains limited (as shown in the table below). This is also consistent with several prior works such as DAPO [6], SkyWork-OR1 [8], and Dr.GRPO [9], which remove the KL term in order to preserve final performance.
>
> |    Method   | AIME24 | AIME25 |  AVG |
> |:-----------:|:------:|:------:|:----:|
> |   KL_0.01   |  46.7  |  33.0  | 39.9 |
> |   KL_0.001  |  47.5  |  32.1  | 39.8 |
> | BAPO (Ours) |  **51.3**  |  **34.6**  | **43.0** |
>
> Second, from the perspective of training efficiency, methods with KL constraints usually require loading an additional model, i.e., a reference model [10], which typically incurs extra GPU memory overhead. For this reason, we did not adopt a KL-based approach.
>
> ---
>
> ### **# Question about entropy collapse**
> Thank you for your insightful comment regarding the relationship between entropy decrease and training instability. We agree that, in the general RL field, a decrease in entropy often indicates that the policy has found a local optimum. However, in language models, as many prior papers have discussed [8][12], entropy collapse usually signals training instability, a loss of exploration and diverse generation ability, and can even lead to failure modes such as repetitive outputs. Therefore, a number of previous works have emphasized stabilizing policy entropy to stabilize the training process [2][3][11].
>
> In addition, as you noted, there are many possible reasons why reinforcement learning can fail for LLMs. We consider entropy to be a particularly important indicator and one of the key research focuses in the LLM RL literature [2][12]. The other factors you mentioned, such as high variance and distribution shift, are also crucial. For example, distribution shift is a classic issue that can be controlled through KL-based constraints; please refer to our previous response (**# Question about policy trust region approaches**). We will discuss these aspects more in future work, as we believe they can provide further insights and inspiration.

---

> ### Author Response · Authors · 2025-11-24
> **Response to Reviewer drrk. Part [2/3]**
>
> ### **# Question about the importance of research questions in our work**
>
> Thank you for your insightful comment.
>
> In the scope of LLMs, algorithms such as PPO and GRPO have been widely proven effective [4][5][13][14]. To control the magnitude of policy updates, these algorithms introduce the clipping trick. Specifically, during policy optimization, clipping constrains the step size of each update, ensuring that the policy does not collapse due to overly aggressive updates.
>
> However, many recent works have investigated the challenges that clipping introduces for LLM RL [6][20-22], and have attempted to address them from both theoretical and empirical perspectives, indicating that this is an active research topic. As you noted, these studies are essentially concerned with trust-region issues in LLM RL, which are indeed important. Our method tackles this challenge from the perspective of clipping and balancing, aiming to mitigate it through balanced positive–negative sampling and entropy stabilization.
>
> ---
>
> ### **# Question about connection to winsorization**
> Thank your very much for your insightful question! We agree that the connection to the classical winsorization literature is important, but we would like to clarify the primary focus of our work. Our goal is not to propose a new, theoretically optimal winsorization scheme for importance sampling in general. Instead, we aim to address a highly practical and acute problem observed in LLM RL training—a problem for which winsorization has also been widely recognized as an effective tool in this domain [6–9]. The issue of asymmetric and heavy-tailed advantages is not merely a theoretical concern; it is a major impediment to stable training in practice. While classic winsorization literature offers many useful ideas, it does not provide a ready-to-use solution for this specific setting, and a full treatment would be beyond the scope of this paper. In future work, we plan to conduct a deeper survey of winsorization-related studies and explore them further. We believe this will be highly insightful and can offer additional guidance for LLM RL training.
>
> ---
>
> ### **# Question about stationary asymmetric clip constant v.s. dynamic clipping**
>
> We appreciate the reviewer’s keen observation regarding the stability of clip bounds in Figure 8. While a stationary asymmetric clip might seem sufficient based on the visual trend in that specific figure, we advocate for the dynamic BAPO approach for several primary reasons:
>
> **Superior performance and entropy stability:** To directly address the reviewer’s hypothesis, we empirically compared BAPO against a "well-tuned" stationary asymmetric clip (fixed at the average values from Figure 8). The results (detailed in **General Response Point #3**) show that the stationary approach consistently underperforms. Crucially, we found that static bounds may fail to prevent entropy collapse or unexpected spikes in policy updates, which was also detailed in [3]'s Ablation Experiments 10. In contrast, BAPO’s step-wise adjustment acts as a stabilizer, maintaining the delicate balance of entropy required for sustained learning.
>
> **Adaptability across civerse training dynamics:** The stability observed in Figure 8 is specific to that particular model/dataset configuration. However, for more challenging tasks or weaker models, the optimal clipping bounds are **not stationary**. As illustrated in our additional results (see **Figure 14 in Appendix C**, training dynamics of DeepSeek-R1-Distill-Qwen-7B on General Reasoner), the clip bounds exhibit a clear evolving trend: they start high to compensate for sparse positive signals during early exploration and gradually decrease as the model converges. A stationary clip cannot accommodate these shifting phases—it would either be too restrictive at the start (hindering exploration) or too loose at the end (destabilizing convergence). BAPO handles this shift automatically.
>
> **Automated tuning vs. manual search:** Regarding the concern about computing the constant at every step: the computational overhead of BAPO’s calculation is mathematically negligible compared to the forward/backward passes of LLM training(which is also detailed in the response to reviewer Zvwr **# Question about computational overhead**). In contrast, finding the "perfect" stationary asymmetric bounds for a new dataset or model requires extensive, costly grid search experiments. BAPO eliminates this manual search cost by automatically identifying the optimal bounds on the fly, making it a much more practical and robust choice for practitioners.
>
> In summary, while the bounds may appear stable in some converged states, the dynamic mechanism provides essential robustness across different training stages and tasks that a static value cannot replicate.

---

> ### Author Response · Authors · 2025-11-24
> **Response to Reviewer drrk. Part [3/3]**
>
> ### **# Question about $c_{low}=0$ in Eq. (8)**
>
> Thank you for your question. In Eq. (8), instead of setting $c_{\text{low}}=0$, we clarify that any value $x<c_{\text{high}}$ can be used here, since $c_{\text{low}}$ has no influence on positive tokens. Specifically, this is due to the $\min$ operation in the loss and the fact that the advantage $A$ for positive tokens satisfies $A>0$:
>
> - When $x < r < c_{\text{high}}$, we have $\text{clip}(r, x, c_{\text{high}}) = r$, and thus $\min(r \cdot A, \text{clip}(r, x, c_{\text{high}}) \cdot A) = r \cdot A$;
>
> - When $r < x$, we have $\text{clip}(r, x, c_{\text{high}}) = x$, and still $\min(r \cdot A, clip(r, x, c_{\text{high}}) \cdot A) = r \cdot A$.
>
> Thus, for any $x < c_{\text{high}}$, $\min(r \cdot A, \text{clip}(r, x, c_{\text{high}}) \cdot A) = r \cdot A$, which is independent of $x$. To highlight this, we simply take $x=0$ here (noting that, by definition, $r \ge 0$).
>
> ---
>
> We once again thank you for your valuable feedback! If you have any further questions, please feel free to let us know and we will do our utmost to respond. If you find our replies satisfactory, we kindly ask that you consider updating your score and confidence accordingly.
>
> ---
>
> **References**
>
> [1] A Minimalist Approach to LLM Reasoning: from Rejection Sampling to Reinforce.
>
> [2] The Entropy Mechanism of Reinforcement Learning for Reasoning Language Models.
>
> [3] Rethinking Entropy Interventions in RLVR: An Entropy Change Perspective.
>
> [4] Training language models to follow instructions with human feedback.
>
> [5] DeepSeekMath: Pushing the Limits of Mathematical Reasoning in Open Language Models.
>
> [6] DAPO: an Open-Source LLM Reinforcement Learning System at Scale.
>
> [7] VAPO: Efficient and Reliable Reinforcement Learning for Advanced Reasoning Tasks.
>
> [8] Skywork Open Reasoner 1 Technical Report.
>
> [9] Understanding R1-Zero-Like Training: A Critical Perspective.
>
> [10] Secrets of RLHF in Large Language Models Part I: PPO.
>
> [11] Dual-Token Constraints for RLVR.
>
> [12] Beyond the 80/20 Rule: High-Entropy Minority Tokens Drive Effective Reinforcement Learning for LLM Reasoning.
>
> [13] DeepSeek-R1: Incentivizing Reasoning Capability in LLMs via Reinforcement Learning.
>
> [14] Qwen3 Technical Report.
>
> [15] MiniMax-M1: Scaling Test-Time Compute Efficiently with Lightning Attention.
>
> [16] Group Sequence Policy Optimization.
>
> [17] UloRL:An Ultra-Long Output Reinforcement Learning Approach for Advancing Large Language Models' Reasoning Abilities.
>
> [18] TAPERED OFF-POLICY REINFORCE Stable and efficient reinforcement learning for LLMs.
>
> [19] Training Large Language Models for Reasoning through Reverse Curriculum Reinforcement Learning.

---

> ### Author Response · Authors · 2025-11-27
> **Invitation to discussion**
>
> Dear Reviewer drrk,
>
> We would like to thank you for the thoughtful and constructive feedback, and appreciate that you agree on the strengths of our paper. During the rebuttal, we performed additional experiments and provided detailed clarifications to address your concerns. The modifications and analyses in our response are summarized as follows:
>
> - We conducted comparative experiments and added clarifications to illustrate why BAPO actively maintains a proportion of positive samples.
>
> - We conducted experiments on policy trust region approaches to validate their feasibility.
>
> - We added clarifications regarding the relationship between entropy decrease and training instability in language models.
>
> - We added clarifications to emphasize the importance of research questions in our work.
>
> - We added clarifications regarding the connection to winsorization.
>
> - We conducted experiments using a fixed asymmetric clip to empirically validate the necessity of dynamic adjustment.
>
> - We added detailed response to your questions about equations.
>
> We are wondering if the responses have addressed your concerns and would appreciate it if you consider raising your final rating. Looking forward to your further feedback!
>
> Best Regards,
>
> Authors of ICLR 2026 Conference Paper 20329

---

> > ### Comment · Reviewer_drrk · 2025-11-27
> >
> > I thank the authors for their additional experiments and responses. However, from experiments alone, most of my fundamental concerns for this paper remain unsolved. I acknowledge that the empirical results reported look good, but it feels more like just a good technical report. The core algorithmic contribution in adaptive clipping, given that the RL community has been using it for sparse reward problems for some time, is somewhat vague to me. Conceptually, the discussion and arguments specific to LLM settings are mostly based on empirics, which are not fundamentally grounded enough. I decide to maintain my score.

---

> > > ### Author Response · Authors · 2025-11-30
> > >
> > > Thank you for your recognition of our empirical results. However, we would like to clarify the following points:
> > >
> > > Our work **primarily lies in LLM RL domain**, rather than traditional RL. Moreover, we want to clarify that the problems we try to address are widely presented and studied in the LLM RL community [1-6], revealing that the research challenges are important.
> > >
> > > We think that the contributions of our work are not limited to a technical report. Instead, through both theoretical analysis and empirical investigation, we uncovered the imbalanced optimization and the Entropy-Clip Rule under the instability in off-policy RL for LLMs (See Section 3 and Appendix G for detailed analysis and theoretical proof). We believe this offers important theoretical contributions and insights to the community.
> > >
> > > Together, these insights collectively form our motivation and represent what we consider our key contributions in both theory and analysis. Further, we proposed BAPO, a method contribution, to address the challenges and finally achieve strong experimental results.
> > >
> > > Thank you for your feedback again, and it has significantly improved our work. We will improve our manuscript accordingly!
> > >
> > > **References**
> > >
> > > [1] The Entropy Mechanism of Reinforcement Learning for Reasoning Language Models.
> > >
> > > [2] Rethinking Entropy Interventions in RLVR: An Entropy Change Perspective.
> > >
> > > [3] DAPO: an Open-Source LLM Reinforcement Learning System at Scale.
> > >
> > > [4] Dual-Token Constraints for RLVR.
> > >
> > > [5] Beyond the 80/20 Rule: High-Entropy Minority Tokens Drive Effective Reinforcement Learning for LLM Reasoning.
> > >
> > > [6] TAPERED OFF-POLICY REINFORCE Stable and efficient reinforcement learning for LLMs.

---

### Official Review · Reviewer_CZZd · 2025-10-26

**Soundness:** 3
**Presentation:** 3
**Contribution:** 3
**Rating:** 6
**Confidence:** 4

**Summary:**

The paper proposed a new RL loss for llm training which stablize off policy training

**Strengths:**

* Entropy clipping rule is a good contribution
* the result shows strong improvement

**Weaknesses:**

* hyper parameter complexity
* eval task diversity is very limited (only math)

**Questions:**

For a real LLM RL training, it might need to combine multiple tasks. but they might have different extent of imbalance. How will the algorithm tackle that?

---

> ### Author Response · Authors · 2025-11-24
> **Response to Reviewer CZZd. Part [1/2]**
>
> We appreciate your valuable time, feedback, and for recognizing our contribution (Entropy-Clip Rule) and strong improvement in results. In the following, we will carefully respond to your questions.
>
> ### **# Weakness about hyper parameters**
>
> Thank you very much for your questions and suggestions. Static clip-bound approaches make it difficult to find suitable hyperparameters, and even after fixing the clip bounds, training can still become unstable as it proceeds, leading to suboptimal performance (see **Point #3 of the General Response**).
>
> In contrast, though our method introduces additional hyperparameters, they are mainly used to specify a rough boundary for the clip bounds; the method itself adaptively adjusts the clip bounds, making it more flexible and generally applicable.
>
> Moreover, following your suggestion, we conducted detailed hyperparameter ablations accordingly. We find that our method is not sensitive to these hyperparameters.
>
> (1) Ablation on positive token contribution threshold $\rho_0$:
>
> |      $\boldsymbol{\rho_0}$     | AIME24 | AIME25 | Average |
> |:-----------:|:------:|:------:|:-------:|
> | 0.4 (paper) |  70.8  |  62.5  |   66.7  |
> |     0.45    |  69.6  |  63.8  |   66.7  |
> |     0.5     |  71.3  |  60.8  |   66.1  |
>
> (2) Ablation on movable range of clipping bounds $[a_-, b_-]$ and $[a_+, b_+]$:
>
> |      $\boldsymbol{[a_-, b_-]}$      |      $\boldsymbol{[a_+, b_+]}$      | AIME24 | AIME25 | Average |
> |:------------------:|:------------------:|:------:|:------:|:-------:|
> | [0.6, 0.9] (paper) | [1.2, 3.0] (paper) |  70.8  |  62.5  |   66.7  |
> |     [0.5, 0.9]     |     [1.2, 3.0]     |  70.8  |  60.8  |   65.8  |
> |     [0.6, 0.8]     |     [1.2, 3.0]     |  71.3  |  59.6  |   65.5  |
> |     [0.6, 0.9]     |     [1.0, 3.0]     |  71.3  |  62.5  |   66.9  |
> |     [0.6, 0.9]     |     [1.2, +inf]    |  72.1  |  60.4  |   66.3  |
>
> (3) Ablation on step sizes $\delta_1$ and $\delta_2$:
>
> |    $\boldsymbol{\delta_1}$    |    $\boldsymbol{\delta_2}$    | AIME24 | AIME25 | Average |
> |:------------:|:------------:|:------:|:------:|:-------:|
> | 0.05 (paper) | 0.02 (paper) |  70.8  |  62.5  |   66.7  |
> |     0.02     |     0.02     |  71.3  |  62.5  |   66.9  |
> |     0.05     |     0.05     |  70.8  |  62.5  |   66.7  |
>
> ---
>
> ### **# Weakness about limited eval task diversity**
>
> Thank you for your valuable suggestion regarding the eval task diversity. Following your suggestion, we expanded our evaluation scope to encompass broader domains and widely recognized benchmarks:
>
> - **Domain 1: Mathematical Reasoning.** To verify robustness in mathematical reasoning, we maintained the Skywork Math dataset for training (consistent with the main paper) and tested on three additional benchmarks: **AMC 2023** and **OlympiadBench**.
>
> - **Domain 2: Abstract/Logical Reasoning.** To test generalization on visual and logical pattern recognition, we trained on the **Enigmata** dataset [1] and evaluated on the **ARC-AGI** task.
>
> - **Domain 3: General Disciplinary Reasoning.** To assess capabilities in graduate-level science and diverse subjects, we trained on the **General-Reasoner** dataset [2] and tested on **GPQA-Diamond**.
>
> **Experimental Results:**
>
> As shown in the table below, BAPO outperforms baselines in all three domains.
>
>
> |  Domain |   Training Set   |  Test Set | Original | GRPO | BAPO | Improvement of BAPO |
> |:-----------:|:--------------------:|:-------------:|:------------:|:--------:|:--------:|:-----------------------:|
> |   **Math**  |     skywork Math     |    AMC 2023   |     73.3     |   82.5   | **92.5** |          +19.2          |
> |             |                      | OlympiadBench |     33.6     |   54.2   | **58.4** |          +25.4          |
> | **Logical** |     Enigmata [1]     |    ARC-AGI    |       0      |    1.4   |  **3.2** |           +3.2          |
> | **General** | General-Reasoner [2] |  GPQA-Diamond |     42.4     |   45.7   | **51.3** |           +8.9          |

---

> ### Author Response · Authors · 2025-11-24
> **Response to Reviewer CZZd. Part [2/2]**
>
> ### **# Question about multi-task training**
>
> Thank you very much for your questions and suggestions—they are very helpful to us. To verify the effectiveness of our method in a multi-task setting, we trained it on multiple tasks jointly, and we did not tune hyperparameters for different tasks/prompts; instead, we used a single unified set of hyperparameters throughout. Our experiments were conducted with the DeepSeek-R1-Distill-Qwen-7B model, with the maximum sequence length set to 16k.
> Thre results are listed in the following table. We find that our method remains effective under this mixed multi-task training scenario and does not require fine-grained task-specific adjustments.
>
> |    Domain   |    Test Set   | GRPO |   BAPO   | Improvement of BAPO |
> |:-----------:|:-------------:|:----:|:--------:|---------------------|
> |   **Math**  |    AMC 2023   | 85.0 | **92.5** |         +7.5        |
> |             | OlympiadBench | 52.9 | **54.8** |         +1.9        |
> | **Logical** |    ARC-AGI    |  1.4 |  **2.4** |         +1.0        |
> | **General** |  GPQA-Diamond | 45.4 | **50.0** |         +4.6        |
>
> We once again thank you for your valuable comments! If you have any further questions, please feel free to let us know and we will do our utmost to respond. If you find our replies satisfactory, we kindly ask that you consider updating your score and confidence accordingly.
>
> ---
>
> **References**
>
> [1] Enigmata: Scaling Logical Reasoning in Large Language Models with Synthetic Verifiable Puzzles.
>
> [2] General-Reasoner: Advancing LLM Reasoning Across All Domains.

---

> ### Author Response · Authors · 2025-11-27
> **Invitation to discussion**
>
> Dear Reviewer CZZd,
>
> We would like to thank you for your constructive feedback, and appreciate that you agree on the strengths of our paper. During the rebuttal, we performed additional experiments and provided detailed clarifications to address your concerns. The modifications in our response are summarized as follows:
>
> - We conducted ablation studies on key hyperparameters to demonstrate their robustness.
>
> - We expanded our evaluation scope to encompass broader domains (including Mathematical, Abstract/Logical, and General Disciplinary Reasoning) and widely recognized benchmarks.
>
> - We added experiments of training on multiple tasks jointly to verify the effectiveness of our method in a multi-task setting.
>
> We are wondering if the responses have addressed your concerns and would appreciate it if you consider raising your final rating. Looking forward to your further feedback!
>
> Best Regards,
>
> Authors of ICLR 2026 Conference Paper 20329

---

### Official Review · Reviewer_mQ9j · 2025-11-01

**Soundness:** 2
**Presentation:** 2
**Contribution:** 2
**Rating:** 4
**Confidence:** 4

**Summary:**

This paper identifies two issues with standard PPO-like RL algorithms when applied to stale data (e.g., due to sample replay or partial rollouts):
(1) optimization imbalance, where negative-advantage samples dominate the policy gradient;
(2) systematic suppression of entropy-increasing updates, which limits exploration.
To address these issues, the authors propose BAPO, a method that adaptively adjusts the clipping range to ensure that the proportion of loss contributed by positive-advantage samples remains above a pre-specified threshold.
Empirical results indicate that BAPO-trained models outperform many existing models on the AIME 2024/2025 benchmarks.

**Strengths:**

* This work provides a sensible and relevant analysis of issues commonly encountered in practice.
* The proposed method is reasonable, practical, and straightforward to implement.
* Strong model performance on AIME 2024 / 2025 benchmarks.

**Weaknesses:**

- **Limited evaluation scope.**
Evaluation are restricted to AIME 2024/2025, each containing only 30 problems. Broader evaluation on more diverse and widely used benchmarks would strengthen the claims.

- **Attribution of improvements is unclear.**
Comparisons are made with prior models trained under substantially different setups, notably in terms of training data. This paper lacks detail on the SFT stage (before RL), making it difficult to determine whether performance gains of the final models stem primarily from BAPO or from the SFT data. Surprisingly, the SFT models (BP-Math-7B/32B-SFT) already outperform baseline models in most cases, as shown in Table 1, suggesting that SFT data might play a major role.

- **Hyperparameter tuning.**
BAPO introduces 7 new hyperparameters (Line 272). The authors claim these are not finely tuned and that BAPO reduces manual tuning compared to methods like DAPO (for which the upper and lower clip ranges are tunable).
However, two different hyperparameter sets are used for the main experiments in Section 5 and the additional experiments in Appendix B, suggesting that some setting-specific tuning is still required.
Moreover, Figure 8 shows that the adaptive clip range (determined by BAPO) remains rather stable throughout, raising the question of whether well-tuned GRPO/DAPO could achieve comparable results.

- **Potential exaggeration of instability issues.**
Some claims and empirical results may overstate the instability caused by data staleness or off-policy RL. In fact, splitting a batch into 4 or 8 mini-batches (corresponding to staleness 4 or 8, if I understand correctly) has been common practice and worked well with standard GRPO/DAPO algorithms in broad settings.

- **Clarify issues about the analysis.**
See questions below.

**Questions:**

Questions about Eq. (5), which rewrites the standard PPO-style policy gradient:
* For both terms on the right-hand side, should the leading $\pi_{\theta}(y_t)$ terms be replaced with the probability ratio $r_t$?
If so, how does this affect the theoretical analysis about entropy later in this work, e.g., Eq. (6)?
* On the left-hand side, please make explicit that $\nabla J^{PPO}$ is defined for a specific $x, y_t$, in contrast to the loss $J^{PPO}$ in Eq. (4).


Questions about Eq. (8):
* Why does the left-hand side contain $\pi_{rollout}$ terms?
* Is there a guarantee that the numerator should be smaller than the denominator?
Note that the denominator sums both positive terms (same as numerator) and negative terms before taking the absolute value.

---

> ### Author Response · Authors · 2025-11-24
> **Response to Reviewer mQ9j. Part [1/4]**
>
> We appreciate your valuable time, insights, and for highlighting our strengths (e.g., **the reasonable and practical method, and the strong model performance**). In the following, we will carefully respond to your questions. We hope we can address your concerns.
>
> ### **# Weakness about limited evaluation scope**
> Thank you for your valuable suggestion regarding the broader evaluation. Following your suggestion, we have significantly expanded our evaluation scope to encompass broader domains and widely recognized benchmarks:
>
> - **Domain 1: Mathematical Reasoning.** To verify robustness in mathematical reasoning, we maintained the Skywork Math dataset for training (consistent with the main paper) and tested on three additional benchmarks: **AMC 2023** and **OlympiadBench**.
>
> - **Domain 2: Abstract/Logical Reasoning.** To test generalization on visual and logical pattern recognition, we trained on the **Enigmata** dataset [1] and evaluated on the **ARC-AGI** task.
>
> - **Domain 3: General Disciplinary Reasoning.** To assess capabilities in graduate-level science and diverse subjects, we trained on the **General-Reasoner** dataset [2] and tested on **GPQA-Diamond**.
>
> **Experimental Results:**
>
> As shown in the table below, BAPO outperforms baselines in all three domains.
>
>
> |  Domain |   Training Set   |  Test Set | Original | GRPO | BAPO | Improvement of BAPO |
> |:-----------:|:--------------------:|:-------------:|:------------:|:--------:|:--------:|:-----------------------:|
> |   **Math**  |     skywork Math     |    AMC 2023   |     73.3     |   82.5   | **92.5** |          +19.2          |
> |             |                      | OlympiadBench |     33.6     |   54.2   | **58.4** |          +25.4          |
> | **Logical** |     Enigmata [1]     |    ARC-AGI    |       0      |    1.4   |  **3.2** |           +3.2          |
> | **General** | General-Reasoner [2] |  GPQA-Diamond |     42.4     |   45.7   | **51.3** |           +8.9          |
>
> ---
>
> ### **# Weakness about attribution of improvements**
>
> Thank you for raising this point. First, BP-Math-SFT in the manuscript is our internal SOTA model, and we acknowledge that SFT played a fundamental role in establishing its strong baseline performance.
>
> However, a critical challenge motivated this work: when attempting to further optimize this strong model using standard RL algorithms (e.g., **GRPO, DAPO**), we observed **negligible additional gains**, indicating a performance bottleneck. Through a detailed analysis of this limitation, we identified the shortcomings of existing RL methods in this context (i.e., imbalanced optimization and entropy collapse) and proposed **BAPO** as a specific solution. BAPO successfully offers more improvements. Furthermore, BAPO is a widely applicable algorithm, not limited to our internal model. As demonstrated in our manuscript, we validated its effectiveness on diverse open-source architectures, including **DeepSeek-R1-Distill-Qwen** (Figure 11, Line 352) and **OctoThinker-Llama-3B-Long-Zero** (Table 2, Line 448), where it consistently delivered improvements.
>
> To further validate the effectiveness of BAPO, we also conducted experiments between different methods (GRPO, DAPO, TOPR, DCPO, and BAPO) on BP-Math-7B in **Point #5 of the General Response**. The results indicate that our BAPO can achieve significant further improvements on strong base models, demonstrating superior performance compared to other optimization algorithms.

---

> ### Author Response · Authors · 2025-11-24
> **Response to Reviewer mQ9j. Part [2/4]**
>
> ### **# Weakness about hyperparameter tuning**
>
> We appreciate your question and suggestion regarding the hyperparameters and the adaptive mechanism. We would like to clarify the distinction between the **sensitivity** of static clip bounds (in DAPO/GRPO) and the **robustness** of BAPO’s hyperparameters. The core advantage of BAPO is its ability to **dynamically adjust** clip bounds to fit the model's evolving capabilities during training.
>
> - In fixed clipping approaches, the static clip bound is sensitive [3]. Consequently, finding an effective static bound requires extensive, resource-heavy grid search for different tasks.
>
> - In contrast, BAPO adaptively adjusts the clipping range at each step, which reduces the need for manual grid search. The newly introduced hyperparameters are fairly forgiving: they only require specifying a rough range and tend to generalize well. Moreover, we find that the same set of hyperparameters works well across different domains (see Point #1 in the General Response).
>
> Moreover, we conducted ablation studies on the key hyperparameters. As shown below, we can find that performance remains stable across a wide range of values:
>
> (1) Ablation on positive token contribution threshold $\rho_0$:
>
> |      $\boldsymbol{\rho_0}$     | AIME24 | AIME25 | Average |
> |:-----------:|:------:|:------:|:-------:|
> | 0.4 (paper) |  70.8  |  62.5  |   66.7  |
> |     0.45    |  69.6  |  63.8  |   66.7  |
> |     0.5     |  71.3  |  60.8  |   66.1  |
>
> (2) Ablation on movable range of clipping bounds $[a_-, b_-]$ and $[a_+, b_+]$:
>
> |      $\boldsymbol{[a_-, b_-]}$      |      $\boldsymbol{[a_+, b_+]}$      | AIME24 | AIME25 | Average |
> |:------------------:|:------------------:|:------:|:------:|:-------:|
> | [0.6, 0.9] (paper) | [1.2, 3.0] (paper) |  70.8  |  62.5  |   66.7  |
> |     [0.5, 0.9]     |     [1.2, 3.0]     |  70.8  |  60.8  |   65.8  |
> |     [0.6, 0.8]     |     [1.2, 3.0]     |  71.3  |  59.6  |   65.5  |
> |     [0.6, 0.9]     |     [1.0, 3.0]     |  71.3  |  62.5  |   66.9  |
> |     [0.6, 0.9]     |     [1.2, +inf]    |  72.1  |  60.4  |   66.3  |
>
> (3) Ablation on step sizes $\delta_1$ and $\delta_2$:
>
> |    $\boldsymbol{\delta_1}$    |    $\boldsymbol{\delta_2}$    | AIME24 | AIME25 | Average |
> |:------------:|:------------:|:------:|:------:|:-------:|
> | 0.05 (paper) | 0.02 (paper) |  70.8  |  62.5  |   66.7  |
> |     0.02     |     0.02     |  71.3  |  62.5  |   66.9  |
> |     0.05     |     0.05     |  70.8  |  62.5  |   66.7  |
>
> ---
>
> ### **# Weakness about stationary v.s. adaptive clipping**
>
> Thanks for your quesiton! We will present the benefits and rationale for dynamic clipping from two different angles.
>
> 1. Flexibility and Entropy Stability
>
>     The most critical advantage of our Adaptive Clip (BAPO) mechanism is its flexibility. Unlike static approaches, BAPO dynamically adjusts the clipping bounds at each specific training step based on the current model state. This step-wise adjustment is essential for **balancing the training objectives** and, more importantly, **maintaining entropy stability**. In contrast, fixed asymmetric clipping schemes face significant challenges: the hyperparameters are difficult to tune optimally, and a fixed bound often leads to issues where the policy entropy either vanishes or collapses, which was also detailed in [3]'s Ablation Experiments 10. To empirically validate the necessity of dynamic adjustment, we conducted experiments using a fixed asymmetric clip (see **Point #3 in General Response**), where the fixed baseline consistently underperformed compared to BAPO.
>
> 2. Clarification on Figure 8
>
>     Regarding the reviewer's observation that the clip bounds in Figure 8 appear relatively stable, this visual effect is primarily due to the use of batch-level statistics and smoothing for plotting clarity. More importantly, Figure 8 represents only one specific configuration. The necessity of dynamic clipping becomes even more pronounced in challenging scenarios, such as the training dynamics of **DeepSeek-R1-Distill-Qwen-7B on the General Reasoner task (shown in Appendix C Figure 14)**. In this setting, since the data is initially difficult for the model, yielding fewer positive tokens, the algorithm must raise the clip bound to maintain balance; conversely, as the model improves and generates more correct answers in later stages, the clip bound decreases significantly. This dynamic adaptability—adjusting to the model's changing competence—is exactly what a stationary clip method fails to provide.
>
> Thanks again for your question again, which is very helpful for us. We will make the statement more clear and revise the manuscript accordingly.

---

> ### Author Response · Authors · 2025-11-24
> **Response to Reviewer mQ9j. Part [3/4]**
>
> ### **# Weakness about potential exaggeration of instability issues**
>
> We appreciate your feedback! As many previous works have noted, off-policy RL enjoys strong efficiency advantages and is compatible with modern AI infrastructure such as partial rollouts, but it can negatively affect training stability and final performance [3-5]. Therefore, this remains a widely recognized research challenge.
>
> This instability is particularly pronounced in large-scale industrial scenarios. To maximize throughput, practitioners frequently employ aggressive asynchronous techniques that push data staleness to extreme levels. In such regimes, standard methods without specific mitigation strategies are prone to model collapse or divergence. The emergence of specialized work like GSPO [6], CISPO [7], and AREAL [5] further underscores that high-staleness instability is a critical bottleneck rather than a trivial issue. BAPO is specifically designed to address this gap, ensuring robustness not only in mild settings but also in these highly efficient, high-staleness environments where standard baselines fail.
>
> Thanks for your question again. We will revise the manuscript to state this more clearly!
>
> ---
>
> ### **# Questions about Eq. (5) and Eq. (8)**
>
> Thank you for your questions about equations. We try to address your questions one by one:
>
> **1. For questions about Eq. (5)**, which rewrites the standard PPO-style policy gradient:
>
> **(1) For your first question:**
>
> For both terms on the right-hand side, we will provide the derivation process from the objective function (Eq. 4) to the gradient formula (Eq. 5) to demonstrate that it should be $\pi_{\theta}(y_t)$ instead of $r_t$ here:
>
> Normally, the PPO surrogate objective is widely used:
>
> $$J^{\text{PPO}}(\theta)=\mathbb{E}_ {x \sim \mathcal{D},\, y \sim \pi_{\text{rollout}}(\cdot|x)} \left[ \sum_{t=1}^{T} \min\big( r_t \cdot A_t, \mathrm{clip}(r_t,1-\varepsilon,1+\varepsilon) \cdot A_t \big) \right],$$
>
> Fixing a particular time step $t$, we can rewrite:
>
> $$J^{\text{PPO}}(\theta)=\sum_{x,y_t} \pi_{\text{rollout}}(y_t\mid x,y_{<t}) \cdot \min \big(r_t \cdot A_t,\mathrm{clip}(r_t,1-\varepsilon,1+\varepsilon) \cdot A_t\big).$$
>
> Note that when $r_t$ lies within the clipping interval, $\mathrm{clip}(r_t,\cdot,\cdot)=r_t$, and thus the two terms are equal.
>
> When $r_t$ exceeds the interval, the second term becomes a constant, resulting in zero gradient with respect to $\theta$.
>
> Meanwhile, observe that $r_t \cdot \pi_{\text{rollout}}=\pi_\theta$.
>
> Therefore, we can express the objective in a piecewise form based on the sign of $A_t$:
>
> $$J^{\text{PPO}}(\theta)=\sum_{x,y_t} \Big[ \mathbb{I} \textbraceleft A_t>0 \textbraceright  \cdot \mathbb{I} \textbraceleft r_t\le1+\varepsilon \textbraceright  \cdot \pi_\theta(y_t\mid x,y_{<t}) \cdot A_t + \mathbb{I}\textbraceleft A_t<0 \textbraceright  \cdot \mathbb{I}\textbraceleft r_t\ge1-\varepsilon \textbraceright  \cdot \pi_\theta(y_t\mid x,y_{<t}) \cdot A_t \Big].$$
>
> When differentiating with respect to $\theta$, only $\pi_\theta(y_t\mid x,y_{<t})$ depends on $\theta$.
>
> Thus we have:
>
> $$\nabla_\theta J^{\text{PPO}}=\sum_{x,y_t}\Big[\mathbb{I}\textbraceleft A_t>0 \textbraceright  \cdot \mathbb{I}\textbraceleft r_t\le1+\varepsilon \textbraceright   \cdot A_t  \cdot \nabla_\theta \pi_\theta(y_t\mid x,y_{<t})+\mathbb{I}\textbraceleft A_t<0 \textbraceright  \cdot \mathbb{I}\textbraceleft r_t\ge1-\varepsilon \textbraceright  \cdot A_t \cdot \nabla_\theta \pi_\theta(y_t\mid x,y_{<t})\Big].$$
>
> Using the log-likelihood identity $\nabla_\theta \pi_\theta = \pi_\theta \nabla_\theta \log \pi_\theta$, the above becomes:
>
> $$\nabla_\theta J^{\text{PPO}}=\sum_{x,y_t} \Big[\mathbb{I}\textbraceleft A_t>0 \textbraceright  \cdot \mathbb{I}\textbraceleft r_t\le1+\varepsilon \textbraceright  \cdot \pi_\theta(y_t\mid x,y_{<t}) \cdot A_t \cdot \nabla_\theta \log \pi_\theta(y_t\mid x,y_{<t})+\mathbb{I}\textbraceleft A_t<0 \textbraceright  \cdot \mathbb{I}\textbraceleft r_t\ge1-\varepsilon \textbraceright  \cdot \pi_\theta(y_t\mid x,y_{<t}) \cdot A_t \cdot \nabla_\theta \log \pi_\theta(y_t\mid x,y_{<t}) \Big].$$
>
> **(2) For your second question:**
>
> In Eq. (5), $\nabla J^{PPO}$ is defined for a specific $x$, $y_t$, which does not align with Eq. (4). We sincerely apologize for this oversight and greatly appreciate you pointing it out. The correction has been made in the latest version of our manuscript.

---

> ### Author Response · Authors · 2025-11-24
> **Response to Reviewer mQ9j. Part [4/4]**
>
> **2. For questions about Eq. (8):**
>
> **(1) For your first question:**
>
> For $\pi_{\text{rollout}}$ in left-hand side: Our objective is to balance contributions of positive and negative sample to the loss function. In practice, the contribution to the loss is computed as shown in Eq. (4):
>
> $$\mathbb{E}_ {\boldsymbol{x}\sim\mathcal{D},\ \boldsymbol{y}\sim \pi_{\theta_\text{rollout}}(\cdot|\boldsymbol{x})} \sum_{t=1}^{T} \left[ \min(r_t\cdot A_t, \text{clip}(r_t, 1-\varepsilon, 1+\varepsilon )\cdot A_t)\right].$$
>
> Here, the term $\pi_{\theta_{\text{rollout}}}$ appears inside the expectation. When expanding the expectation into a summation, it is necessary to multiply by the probability distribution, which introduces the $\pi_{\text{rollout}}$ term.
>
> **(2) For your second question:**
>
> For the numerator being smaller than the denominator: We apologize for the misplacement of the absolute value notation in our original manuscript. We sincerely thank you for pointing out this issue, which has allowed us to correct it. The revised Eq. (8) in our latest version is as follows:
>
> $$\frac{\sum_{A_t>0} \pi_{\theta_{\textnormal{rollout}}}(y_t) \cdot |\min(r_t\cdot A_t, \textnormal{clip}(r_t, 0, c_\textnormal{high} )\cdot A_t)|}{\sum_{A_t} \pi_{\theta_{\textnormal{rollout}}}(y_t) \cdot |\min(r_t\cdot A_t, \textnormal{clip}(r_t, c_\textnormal{low}, c_\textnormal{high})\cdot A_t)|} \geq \rho_0.$$
>
> Thus, there is a guarantee qthat the numerator should be smaller than the denominator.
>
>
> We once again thank you for your valuable comments!  If you have any further questions, please feel free to let us know and we will do our utmost to respond. If you find our replies satisfactory, we kindly ask that you consider updating your score and confidence accordingly.
>
> ---
>
> **References**
>
> [1] Enigmata: Scaling Logical Reasoning in Large Language Models with Synthetic Verifiable Puzzles.
>
> [2] General-Reasoner: Advancing LLM Reasoning Across All Domains.
>
> [3] Skywork Open Reasoner 1 Technical Report.
>
> [4] UloRL:An Ultra-Long Output Reinforcement Learning Approach for Advancing Large Language Models' Reasoning Abilities.
>
> [5] AReaL: A Large-Scale Asynchronous Reinforcement Learning System for Language Reasoning.
>
> [6] Qwen3 Technical Report.
>
> [7] MiniMax-M1: Scaling Test-Time Compute Efficiently with Lightning Attention.

---

> ### Author Response · Authors · 2025-11-27
> **Invitation to discussion**
>
> Dear Reviewer mQ9j,
>
> We would like to thank you for the insightful and valuable suggestions, and appreciate that you agree on the strengths of our paper. During the rebuttal, we performed additional experiments and provided detailed clarifications to address your concerns. The modifications and analyses in our response are summarized as follows:
>
> - We expanded our evaluation scope to encompass broader domains (including Mathematical, Abstract/Logical, and General Disciplinary Reasoning) and widely recognized benchmarks.
>
> - We conducted experiments between different methods (GRPO, DAPO, TOPR, DCPO, and BAPO) on BP-Math-7B-SFT model to further validate the effectiveness of BAPO.
>
> - We conducted ablation studies on key hyperparameters to demonstrate their robustness.
>
> - We conducted experiments using a fixed asymmetric clip to empirically validate the necessity of dynamic adjustment.
>
> - We added clarifications on instability issues of off-policy RL.
>
> - We added detailed response to your questions about equations, and corrected the mistakes in the latest version of our manuscript.
>
> We are wondering if the responses have addressed your concerns and would appreciate it if you consider raising your final rating. Looking forward to your further feedback!
>
> Best Regards,
>
> Authors of ICLR 2026 Conference Paper 20329

---

> > ### Comment · Reviewer_mQ9j · 2025-11-27
> >
> > Thank you for the detailed responses. I'll take a closer look at the new experiments later. But for now, I'm still confused about the analysis for the gradient of the PPO objective. When you write "$\sum_{y_t}$", does it mean "$\sum_{y_t \in V}$" where $V$ denotes the vocabulary, or is $y_t$ here a specific sample in the training data?

---

> > > ### Author Response · Authors · 2025-11-29
> > >
> > > Thank you for your question.
> > >
> > > From theoretical perspectives, for a fixed time step t, the $\sum_{y_t}$ means $\sum_{y_t \in V}$, where $V$ denotes the vocabulary. We apologize for not explicitly elaborating on it earlier. If written in a more detailed form, Eq. (5) would be:
> > >
> > > $\nabla_\theta J^{\text{PPO}}=\sum_{x \in \mathcal{D}} \sum_{t=1}^{T} \sum_{y_t \in V} \Big[\mathbb{I}\textbraceleft A_t>0 \textbraceright  \cdot \mathbb{I}\textbraceleft r_t\le1+\varepsilon \textbraceright  \cdot \pi_\theta(y_t\mid x,y_{<t}) \cdot A_t \cdot \nabla_\theta \log \pi_\theta(y_t\mid x,y_{<t})+\mathbb{I}\textbraceleft A_t<0 \textbraceright  \cdot \mathbb{I}\textbraceleft r_t\ge1-\varepsilon \textbraceright  \cdot \pi_\theta(y_t\mid x,y_{<t}) \cdot A_t \cdot \nabla_\theta \log \pi_\theta(y_t\mid x,y_{<t}) \Big].$
> > >
> > > However, during the actual optimization process, considering the entire vocabulary—which typically contains around 50,000 tokens—would lead to unacceptable computational costs. Therefore, in practice, we employ Monte Carlo sampling to approximate the theoretical expectation, optimizing only over the sampled tokens rather than the full vocabulary [1-3]:
> > >
> > > $\nabla_\theta J^{\text{PPO}}=\sum_{x \sim \mathcal{D}, y \sim \pi_{\theta_\text{rollout}(\cdot | x)}} \sum_{t=1}^{T} \Big[\mathbb{I}\textbraceleft A_t>0 \textbraceright  \cdot \mathbb{I}\textbraceleft r_t\le1+\varepsilon \textbraceright  \cdot \pi_\theta(y_t\mid x,y_{<t}) \cdot A_t \cdot \nabla_\theta \log \pi_\theta(y_t\mid x,y_{<t})+\mathbb{I}\textbraceleft A_t<0 \textbraceright  \cdot \mathbb{I}\textbraceleft r_t\ge1-\varepsilon \textbraceright  \cdot \pi_\theta(y_t\mid x,y_{<t}) \cdot A_t \cdot \nabla_\theta \log \pi_\theta(y_t\mid x,y_{<t}) \Big].$
> > >
> > > **References**
> > >
> > > [1] Proximal Policy Optimization Algorithms.
> > >
> > > [2] Trust Region Policy Optimization.
> > >
> > > [3] Secrets of RLHF in Large Language Models Part I: PPO.

---

### Author Response · Authors · 2025-11-24
**General Response. Part [2/2]**

### **#4 Experiments of ablation on hyperparameters**

In response to the questions regarding hyperparameters raised by Reviewers mQ9j and CZZd, we conducted comprehensive ablation studies to demonstrate the robustness of BAPO. The results, **which have also been included in Appendix E of the revised manuscript**, are presented below. As observed, the performance of the BP-Math-7B model remains stable across a wide range of values.

(1) Ablation on positive token contribution threshold $\rho_0$:

|      $\boldsymbol{\rho_0}$     | AIME24 | AIME25 | Average |
|:-----------:|:------:|:------:|:-------:|
| 0.4 (paper) |  70.8  |  62.5  |   66.7  |
|     0.45    |  69.6  |  63.8  |   66.7  |
|     0.5     |  71.3  |  60.8  |   66.1  |

(2) Ablation on movable range of clipping bounds $[a_-, b_-]$ and $[a_+, b_+]$:

|      $\boldsymbol{[a_-, b_-]}$      |      $\boldsymbol{[a_+, b_+]}$      | AIME24 | AIME25 | Average |
|:------------------:|:------------------:|:------:|:------:|:-------:|
| [0.6, 0.9] (paper) | [1.2, 3.0] (paper) |  70.8  |  62.5  |   66.7  |
|     [0.5, 0.9]     |     [1.2, 3.0]     |  70.8  |  60.8  |   65.8  |
|     [0.6, 0.8]     |     [1.2, 3.0]     |  71.3  |  59.6  |   65.5  |
|     [0.6, 0.9]     |     [1.0, 3.0]     |  71.3  |  62.5  |   66.9  |
|     [0.6, 0.9]     |     [1.2, +inf]    |  72.1  |  60.4  |   66.3  |

(3) Ablation on step sizes $\delta_1$ and $\delta_2$:

|    $\boldsymbol{\delta_1}$    |    $\boldsymbol{\delta_2}$    | AIME24 | AIME25 | Average |
|:------------:|:------------:|:------:|:------:|:-------:|
| 0.05 (paper) | 0.02 (paper) |  70.8  |  62.5  |   66.7  |
|     0.02     |     0.02     |  71.3  |  62.5  |   66.9  |
|     0.05     |     0.05     |  70.8  |  62.5  |   66.7  |

The experimental results show that our method is not highly sensitive to hyperparameters. Its key advantage is that it can automatically find appropriate clipping values within a broad range, even though we still need to specify the upper and lower bounds.

---

### **#5 Experiments of more baselines for fixed original model**

Following Reviewer Zvwr's suggestion, we conducted a systematic comparison between different methods (TOPR [6], DAPO [3], GRPO [7], and DCPO [8]) using the same backbone model (BP-Math-7B). The results are shown in the table below and **have also been added to Section 5.3 (Discussion) of the revision**.

|             |  AIME24  |  AIME25  |    AVG   |
|:-----------:|:--------:|:--------:|:--------:|
|     SFT     |   66.9   |    59    |   62.9   |
|     GRPO    |   69.2   |   59.2   |   64.2   |
|     DAPO    |   68.3   |   60.8   |   64.6   |
|     TOPR    |   68.8   |   60.4   |   64.6   |
|     DCPO    |   68.8   |   57.9   |   63.4   |
| BAPO (Ours) | **70.8** | **62.5** | **66.7** |

It can be observed that BAPO consistently outperforms these strong baselines.

---

**References**

[1] Enigmata: Scaling Logical Reasoning in Large Language Models with Synthetic Verifiable Puzzles.

[2] General-Reasoner: Advancing LLM Reasoning Across All Domains.

[3] DAPO: An Open-Source LLM Reinforcement Learning System at Scale.

[4] Skywork Open Reasoner 1 Technical Report.

[5] Understanding R1-Zero-Like Training: A Critical Perspective.

[6] Trust Region Policy Optimization.

[7] DeepSeekMath: Pushing the Limits of Mathematical Reasoning in Open Language Models.

[8] DCPO: Dynamic Clipping Policy Optimization.

---

### Author Response · Authors · 2025-11-24
**General Response. Part [1/2]**

Dear reviewers, we sincerely thank you for your valuable feedback. Your insightful suggestions have greatly improved our paper. We appreciate the recognition of our strengths, particularly **the insightful motivation [Reviewer mQ9j, drrk, and Zvwr], the reasonable and practical method BAPO [Reviewer mQ9j and Zvwr], the contribution of Entropy-Clip Rule [Reviewer CZZd], and the strong model performance [Review mQ9j, CZZd, and Zvwr]**.

Following your suggestions, we have conducted additional experiments and provided clarifications on some common issues, revising the manuscript accordingly. Specifically, we will provide the following responses in the hope of resolving your concerns.

---
### **#1 Experiments of broader evaluation scope**

Following the suggestions of Reviewers mQ9j and CZZd, we expanded our evaluation scope to cover broader domains and widely recognized benchmarks, namely AMC 2023 and OlympiadBench for mathematical reasoning, ARC-AGI for logical reasoning, and GPQA-Diamond for science question answering. The experiments were conducted on DeepSeek-R1-Distill-Qwen-7B. For the math reasoning and science QA tasks, we used a maximum sequence length of 8k. For logical reasoning, we adopted a 16k sequence length, since the questions typically include few-shot examples and are therefore relatively long.

The experimental results are presented in the table below. We have also **incorporated these findings into Section 5.3 (Discussion)** of the revised manuscript. We can find that BAPO brings substantial improvements, validating its generality across different domains.

|  Domain |   Training Set   |  Test Set | Original | GRPO | BAPO | Improvement of BAPO |
|:-----------:|:--------------------:|:-------------:|:------------:|:--------:|:--------:|:-----------------------:|
|   **Math**  |     skywork Math     |    AMC 2023   |     73.3     |   82.5   | **92.5** |          +19.2          |
|             |                      | OlympiadBench |     33.6     |   54.2   | **58.4** |          +25.4          |
| **Logical** |     Enigmata [1]     |    ARC-AGI    |       0      |    1.4   |  **3.2** |           +3.2          |
| **General** | General-Reasoner [2] |  GPQA-Diamond |     42.4     |   45.7   | **51.3** |           +8.9          |

---

### **#2 Experiments with policy turst region approaches**

Reviewer drrk’s question mentioned trust-region approaches, such as KL-based policy gradient methods, which can improve training stability and mitigate entropy collapse. Following this suggestion, we conducted experiments on the DeepSeek-R1-Distill-Qwen-7B model with max-length of 8k. We found that this strategy does provide some improvement on training stability; however, the final model performance is limited (as shown in the table below). This observation is also consistent with many previous works like DAPO [3], SkyWork-OR1 [4], and Dr.GRPO [5], which remove the KL term to preserve final performance.

|    Method   | AIME24 | AIME25 |  AVG |
|:-----------:|:------:|:------:|:----:|
|   KL_0.01   |  46.7  |  33.0  | 39.9 |
|   KL_0.001  |  47.5  |  32.1  | 39.8 |
| BAPO (Ours) |  **51.3**  |  **34.6**  | **43.0** |

---

### **#3 Experiments of varying static clip bounds**

Following the suggestions of Reviewers mQ9j and drrk, we conducted experiments on the DeepSeek-R1-Distill-Qwen-7B model to examine whether fixing the clipping threshold within a bounded range can yield better performance. Through the detailed ablations in the following table, we found that these fixed-threshold variants cannot surpass our method. Meanwhile, we observed that the fixed asymmetric strategy is still prone to training instability or entropy collapse. In other words, as training progresses over time, a fixed asymmetric clipping setting struggles to adaptively balance positive and negative samples and to stabilize entropy. This further corroborates the advantage of our approach, which adaptively adjusts the clipping threshold at each step.

|       Method       |  AIME24  |  AIME25  |    AVG   |
|:------------------:|:--------:|:--------:|:--------:|
| Clip w/ [0.8, 1.5] |   49.2   |   32.5   |   40.9   |
| Clip w/ [0.8, 2.0] |   48.8   |   31.7   |   40.3   |
| Clip w/ [0.8, 2.5] |   45.4   |   33.0   |   39.2   |
| Clip w/ [0.9, 2.0] |   50.4   |   32.1   |   41.3   |
| Clip w/ [0.7, 2.0] |   49.6   |   32.1   |   40.9   |
|     BAPO (Ours)    | **51.3** | **34.6** | **43.0** |

---

### Author Response · Authors · 2025-12-02
**Summary**

We sincerely thank all the reviewers, the Area Chair, the Senior Area Chairs and Program Chairs for the valuable feedback and time. We also deeply value thoughtful evaluations, which highlight the strengths of our work:
1. Insightful motivation with a sensible and relevant analysis of issues commonly encountered in practice. [Reviewer mQ9j, drrk and Zvwr]
2. Theoretical contribution of introducing the Entropy-Clip Rule and in-depth insights into training dynamics of LLM RL. [Reviewer CZZd and Zvwr]
3. Effective method BAPO, which is flexible, reasonable, practical, and easy to implement. [Reviewer mQ9j and Zvwr]
4. Strong performance across different off-policy settings, data staleness, model scales, and architectures. [Reviewer mQ9j, CZZd, drrk, and Zvwr]
5. Well-written, effectively highlighted the key research points, and is easy to follow. [Reviewer drrk and Zvwr]

A major concern shared by reviewers is that the original evaluation is restricted to mathematical reasoning tasks (AIME24 and AIME25). Another concern is the introduction of additional hyperparameters, which they think may require careful tuning. A common suggestion across reviewers is to compare BAPO with a broader range of baselines, specifically algorithms like TOPR, DAPO, and DCPO. Reviewers also questioned whether trust-region approaches can effectively address instability in off-policy RL.

In our rebuttal, we conducted additional experiments and offered clarifications to respond to these suggestions and questions, including:
- Expanded evaluation to more diverse and widely-used tasks and benchmarks, including mathematical (AMC 2023, OlympiadBench), abstract/logical (ARC-AGI), and general disciplinary reasoning (GPQA-Diamond) [mQ9j, CZZd]
- Compared with trust-region approaches, and more baseline methods like TOPR, DAPO, and DCPO [drrk, Zvwr]
- Conducted experiments of varying static clip bounds to validate the necessity of dynamic adjustment [mQ9j, drrk]
- Performed comprehensive ablation studies on key hyperparameters [mQ9j, CZZd]
- Added experiments of training on multiple tasks jointly to verify the applicability of BAPO on such scenarios without detailed and careful design [CZZd]
- Added detailed responses regarding the equations and derivations [mQ9j, drrk, Zvwr]
  - Provided the derivation process from the objective function (Eq. 4) to the gradient formula (Eq. 5) [mQ9j]
  - Added detailed explanation and rewrote Eq. (5) and Eq. (8) [mQ9j]
  - Added detailed explanation on why $c_{\text{low}}$ has no influence on positive tokens [drrk]
  - Provided a more precise definition of advantage $A_t$ and $z_{y,x}$ [Zvwr]

The above analysis and experiments are consistently promising, offering additional validation of the BAPO method's effectiveness and generalizability. The clarifications also help improve the quality of the manuscript.

During discussion period, we received responses from Reviewer drrk, who thought our work feels like "a good technical report". However, we think our contributions are multifaceted, spanning analytical, theoretical, and methodological dimensions. Other reviewers also highlighted our theoretical contributions. For instance, Reviewer mQ9j pointed out that our work "provides a sensible and relevant analysis", Reviewer CZZd noted that "Entropy clipping rule is a good contribution", and Reviewer Zvwr also emphasized our "theoretical insights".

Reviewer mQ9j, CZZd, and Zvwr didn't participate in the discussion, but we have made every effort to address their concerns through detailed responses and additional experiments. We believe this can effectively resolve their key concerns.

We have revised our manuscript based on the rebuttal content to further strengthen our work. Finally, we greatly appreciate the reviewers' recognition and their constructive suggestions. We hope our work can provide insights for the LLM RL community.

Best regards,

Authors of ICLR 2026 Conference Paper 20329

---

### Meta-Review · Area_Chair_gwZS · 2026-01-06

**Summary:**

The reviewers initially questioned the evaluation task complexity, the sensitivity of hyperparameters, and most importantly, the target issue itself. For example, one of the reviewers mentioned the issue might be artificial or somewhat exaggerated. However, most of the concerns are addressed in their detailed rebuttal text.

**Reviewer Concerns:**

Most of the concerns are addressed.

**Reviewer Scores:**

6666

---

### Decision · Program_Chairs · 2026-01-26

Accept (Poster)